# Exploring Structured Semantic Priors Underlying Diffusion Score for Test-time Adaptation

**Mingjia Li**
Beijing Institute of Technology
mingjiali@bit.edu.cn

**Shuang Li**[*]
Beihang University
shuangliai@buaa.edu.cn

**Tongrui Su**
Beijing Institute of Technology
molarsu@bit.edu.cn

**Longhui Yuan**
Beijing Institute of Technology
longhuiyuan@bit.edu.cn

**Jian Liang**
Kuaishou Technology
liangjian03@kuaishou.com

**Wei Li**[*]
Inceptio Technology
liweimcc@gmail.com

## Abstract

Capitalizing on the complementary advantages of generative and discriminative models has always been a compelling vision in machine learning, backed by a growing body of research. This work discloses the hidden semantic structure within score-based generative models, unveiling their potential as effective discriminative priors. Inspired by our theoretical findings, we propose DUSA to exploit the structured semantic priors underlying diffusion score to facilitate the test-time adaptation of image classifiers or dense predictors. Notably, DUSA extracts knowledge from a single timestep of denoising diffusion, lifting the curse of Monte Carlo-based likelihood estimation over timesteps. We demonstrate the efficacy of our DUSA in adapting a wide variety of competitive pre-trained discriminative models on diverse test-time scenarios. Additionally, a thorough ablation study is conducted to dissect the pivotal elements in DUSA. Code is publicly available at https://github.com/BIT-DA/DUSA.

## 1 Introduction

The combination of generative and discriminative modeling has always been appealing due to their distinct nature in data comprehension [1, 2, 3, 4]. Discriminative models are adept at making accurate predictions on training data [5, 6, 7, 8, 9, 10, 11], but can be fragile when confronted with unseen data [12]. This vulnerability can be attributed to their tendency to learn spurious correlation as a shortcut, hindering their transferability [13]. Generative models, however, are proficient in capturing the underlying structure of the data, giving them an edge in grasping the whole picture [14, 15, 16] and enhancing robustness [17, 18]. Prior works have verified the effectiveness of generative objectives in discriminative learning [4, 19], yet the utilization of pre-trained generative models is under-explored.

The recent surge of diffusion models [20, 21, 22] has ignited interest in adopting them for applications beyond image generation [23, 24, 25, 26]. In the context of test-time adaptation, a pre-trained task model is updated on the fly to make accurate predictions on incoming target samples without access to their labels. This presents challenges, as the target data distribution may differ from that encountered during pre-training. The literature reveals that we can not only extract discriminative features from capacious diffusion models [23, 24, 26, 27, 28], but also convert these models into generative classifiers that demonstrate human-level generalization on out-of-distribution samples [25, 29, 30]. Such properties render them viable choices for facilitating the test-time adaptation of discriminative models, which may underperform on unseen data [31]. Diffusion-TTA [32] ranks among the first

---

[*]Corresponding author.

38th Conference on Neural Information Processing Systems (NeurIPS 2024).

to employ diffusion models for test-time adaptation, where task outputs are used to modulate the conditioning of a diffusion model with the objective of likelihood maximization. While Diffusion-TTA is competitive, there's still room to unleash the full potential of diffusion models. To achieve this, two key aspects warrant further exploration. First, the conditioning space is typically low-dimensional in diffusion models [20, 22], which restricts its ability to capture the intricacies of complex data and thus impedes its expressiveness as a discriminative prior. Further, the common image-level condition lacks a fine-grained connection with data, limiting its potential in guiding dense prediction tasks. Conversely, the high-dimensional latent space of diffusion models exhibits a surprisingly interpretable semantic structure [33, 34, 35, 36, 37], making it a good fit for assisting discriminative tasks and easily extensible to dense prediction. Second, Diffusion-TTA is heavily reliant on the Monte Carlo method over as many as 180 timesteps to estimate a biased approximation of likelihood [32, 38], resulting in high computational complexity proportional to sampled timesteps.

With this work, we aim to boost test-time adaptation performance by digging into the semantic structure of diffusion models in the latent space, while lifting reliance on the Monte Carlo sampling of timesteps. Although a few works elucidate the semantic properties of the latent space [34, 37], they all take a generative viewpoint and are not tailored for discriminative tasks. We instead depart from the perspective of score functions [39, 40, 41] on the latent space, which is closely related to the denoising diffusion formulation [38, 42, 43]. Our method features exploring the structured semantic priors underlying **D**iff**U**sion models as **S**core estimators for test-time **A**daptation (**DUSA**).

Concretely, we start by providing a theoretical illustration of the semantic structure underneath the score functions $\nabla_{\mathbf{x}} \log p(\mathbf{x} \mid y)$, where the conditional probability $p(y \mid \mathbf{x})$ is implicitly embedded. The theoretical findings not only unveil discriminative priors hidden within score-based diffusion models, but applies to every single timestep and avoids likelihood estimation. A test-time objective is then derived by substituting the pre-trained task model and diffusion model for the implicit priors and score functions, respectively. Intuitively, the precise score estimation by diffusion models forms a well-structured semantic space, where the task model can learn implicit discriminative priors. Given their generative nature, the priors are further blessed with improved robustness, ultimately benefiting task prediction. Another key advantage of our approach lies in shifting computational complexity from timesteps to the number of classes, which aligns closely with our focus on discriminative tasks. Thereby, a more efficient adaptation scheme can be enabled through our practical designs.

Besides, the capacity of our DUSA is testified across a variety of task model families, test-time adaptation protocols, and task categories. Our DUSA consistently outperforms the competitive counterparts in adapting pre-trained classifiers with different backbones to out-of-distribution scenarios, whether in the mild protocol of data from a single domain [44] or the more challenging one with a continually changing datastream [45]. We also showcase the versatility of our DUSA by applying it almost as-is to test-time semantic segmentation. All diffusion models employed are trained on the corresponding source domain of the task model. Extensive analyses of the components in our method back the validity of DUSA and underline the benefits of borrowing knowledge from generative modeling.

Our main contributions can be summarized as:

- A novel proposition is given from a theoretical perspective to extract discriminative priors from score-based diffusion models, which are single-timestep-based and versatile enough to handle both classification and dense prediction tasks at test time.

- Inspired by the proposition and enhanced by practical designs, our DUSA effectively leverages the structured semantic priors and rivals in test-time adaptation with improved efficiency.

- DUSA outperforms the best existing methods by $+5.1\%$ and $+7.3\%$ in fully and continual test-time adaptation on ConvNeXt-L and $+4.2\%$ in test-time semantic segmentation on SegFormer-B5, validating the excellence of our method in extracting valuable priors from diffusion models.

## 2   Preliminaries

**Test-time adaptation.**    A model well-trained on source data can face severe performance degradation on out-of-distribution (OOD) target samples. To tackle this, test-time adaptation (TTA) [44] is proposed to boost model performance at inference time. Formally, an off-the-shelf model $f_\theta(\mathbf{x})$ pre-trained on labeled source data $\mathcal{D}_{\mathcal{S}} = \{(\mathbf{x}_i, y_i)\}_{i=1}^{N}$ is adopted as the task model, where the source data follows a probability distribution $\mathbf{x}_i \sim P_{\mathcal{S}}(\mathbf{x})$ and is inaccessible during adaptation. TTA aims

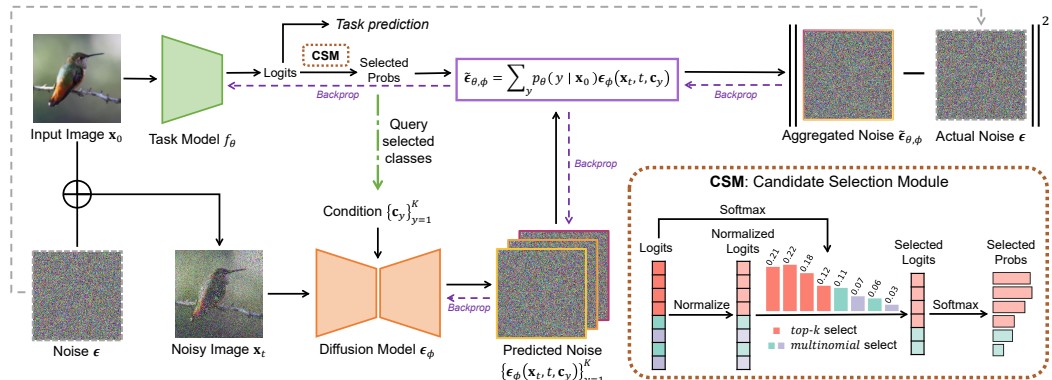

Figure 1: **Overview of DUSA**. Our method adapts a discriminative task model $f_\theta$ with a generative diffusion model $\epsilon_\phi$. Given image $\mathbf{x}_0$ at test-time, the task model outputs logits. To improve efficiency, we devise a CSM to select classes to adapt and return their probabilities (probs). The embeddings of the classes are then queried as diffusion model conditions, yielding conditional noise predictions from noisy image $\mathbf{x}_t$. The aggregated noise $\tilde{\epsilon}_{\theta,\phi}$ is then constructed from ensembling conditional noises with probs, which is aligned with the added noise $\epsilon$ following Eq. (10). Both models are updated.

at pushing the limits of model performance on unlabeled target data $\mathcal{D}_\mathcal{T} = \{\mathbf{x}_j\}_{j=1}^M$ on the fly, where the target data follows $\mathbf{x}_j \sim P_\mathcal{T}(\mathbf{x})$ and $P_\mathcal{S}(\mathbf{x}) \neq P_\mathcal{T}(\mathbf{x})$. With batched target data arriving online, we obtain predictions from the task model $f_\theta(\mathbf{x})$ and update it on live target samples without labels.

**Diffusion models.** Diffusion models excel at modeling data distribution $p(\mathbf{x})$ by learning to restore the gradually destroyed data structure [20, 38, 46]. For diffusion models, a forward and a reverse process is defined. In the forward process, small Gaussian noise is iteratively applied to real data $\mathbf{x}_0$:

$$q(\mathbf{x}_t \mid \mathbf{x}_{t-1}) := \mathcal{N}(\mathbf{x}_t; \sqrt{1 - \beta_t}\mathbf{x}_{t-1}, \beta_t\mathbf{I}), \tag{1}$$

where $\{\beta_t \in (0,1)\}_{t=1}^T$ is a variance schedule defining the noise added at each timestep $t$.

With the reparameterization trick, the noisy version of $\mathbf{x}_0$ can be directly obtained in a single step:

$$\mathbf{x}_t = \sqrt{\bar{\alpha}_t}\mathbf{x}_0 + \sqrt{1 - \bar{\alpha}_t}\epsilon, \quad \epsilon \sim \mathcal{N}(\mathbf{0}, \mathbf{I}), \tag{2}$$

where $\alpha_t := 1 - \beta_t$, $\bar{\alpha}_t := \prod_{i=1}^t \alpha_i$, and the sampled noise $\epsilon$ is of the same dimensionality as $\mathbf{x}_0$.

In the reverse process, a conditional diffusion model [20, 21, 22, 47, 48] $\epsilon_\phi(\mathbf{x}_t, t, \mathbf{c})$ is trained to predict the noise added to $\mathbf{x}_t$ with condition $\mathbf{c}$, by minimizing the simplified denoising objective:

$$\mathcal{L}_{simple}(\phi) := \mathbb{E}_{(\mathbf{x}_0,\mathbf{c}),\epsilon,t}\left[\|\epsilon - \epsilon_\phi(\mathbf{x}_t, t, \mathbf{c})\|_2^2\right]. \tag{3}$$

## 3 Structured Semantic Priors in Diffusion Score for Test-Time Adaptation

In this section, we first review a relevant method (Sec. 3.1), then provide the theoretical insight behind our DUSA (Sec. 3.2). At last, we advocate a few practical designs with efficiency in mind (Sec. 3.3). The framework of our DUSA is illustrated in Fig. 1. Given a pre-trained task model, a set of classes to optimize is selected by the Candidate Selection Module (CSM) based on task model prediction to improve adaptation efficiency. We focus on the selected classes and aggregate the conditional noise estimations with CSM-modulated probabilities on these classes, upon which our DUSA objective is constructed. The structured semantic priors of the diffusion model are then propagated to the task model through our objective. For more details please refer to Alg. 1 in Appendix E.

### 3.1 A Brief Review of Diffusion-TTA

Diffusion-TTA [32] takes the first step in exploring conditional diffusion models for test-time adaptation. In Diffusion-TTA, the task model prediction $p_\theta(y \mid \mathbf{x}_0)$ is integrated with class embeddings

$\{\mathbf{c}_y\}_{y=1}^N$ to get a soft condition $\mathbf{c} = \sum_y p_\theta(y \mid \mathbf{x}_0)\mathbf{c}_y$, where $N$ is the number of classes. The sample-wise adaptation is performed with the diffusion loss: $\mathcal{L}(\theta, \phi) = \mathbb{E}_{t,\boldsymbol{\epsilon}}\big[\|\boldsymbol{\epsilon} - \boldsymbol{\epsilon}_\phi(\mathbf{x}_t, t, \mathbf{c})\|_2^2\big]$. Inspired by [25], the loss is averaged over hundreds of $(t, \boldsymbol{\epsilon})$ pairs to remedy the performance degradation caused by incorrect class prediction using a single timestep [32], at the cost of reduced efficiency.

## 3.2 Unlocking the Discriminative Power of Conditional Diffusion Models

Unlike previous attempts that rely heavily on a massive number of timesteps to provide an appropriate estimation of likelihood $p(\mathbf{x} \mid \mathbf{c})$ [25, 29, 32], we shed light on the semantic structure underneath the denoising capability of conditional diffusion models from the perspective of score functions [39, 40, 41, 42], which will be shown to hold for every single timestep. Proofs can be found in Appendix C.

We first present our main theoretical contribution to reveal the semantic structure of score functions:

**Proposition 1.** *Let $p(\mathbf{x})$ and $\{p(\mathbf{x} \mid y) : y \in \mathcal{Y}\}$ be continuously differentiable probability densities, their score functions $\nabla_\mathbf{x} \log p(\mathbf{x})$ and $\{\nabla_\mathbf{x} \log p(\mathbf{x} \mid y) : y \in \mathcal{Y}\}$, the following equation holds:*

$$\nabla_\mathbf{x} \log p(\mathbf{x}) = \sum_y p(y \mid \mathbf{x})\nabla_\mathbf{x} \log p(\mathbf{x} \mid y). \tag{4}$$

**Remark.** *The equation holds under mild assumptions about the densities but applies to any data $\mathbf{x}$ with an entire set of conditions $\{y : y \in \mathcal{Y}\}$. All score functions can be estimated by score matching or denoising diffusion. Note that the posteriors $\{p(y \mid \mathbf{x}) : y \in \mathcal{Y}\}$ are **not** directly modeled, and thus can be seen as the **implicit priors** hidden in the construction of (conditional) score functions.*

To link our Proposition 1 with a trained diffusion model, we revisit Tweedie's Formula [49], which serves as a key connection between score functions and the formulation of diffusion models [43]:

**Lemma 1** (Tweedie's Formula). *Let $\mathbf{z} \mid \boldsymbol{\mu}_\mathbf{z} \sim \mathcal{N}(\mathbf{z}; \boldsymbol{\mu}_\mathbf{z}, \boldsymbol{\Sigma}_\mathbf{z})$, then the posterior expectation of $\boldsymbol{\mu}_\mathbf{z}$ given $\mathbf{z}$ can be estimated by:*

$$\mathbb{E}[\boldsymbol{\mu}_\mathbf{z} \mid \mathbf{z}] = \mathbf{z} + \boldsymbol{\Sigma}_\mathbf{z}\nabla_\mathbf{z} \log p(\mathbf{z}). \tag{5}$$

As Eq. (2) indicates $q(\mathbf{x}_t \mid \mathbf{x}_0) = \mathcal{N}(\mathbf{x}_t; \sqrt{\bar{\alpha}_t}\mathbf{x}_0, (1 - \bar{\alpha}_t)\mathbf{I})$, we have the following corollary:

**Corollary 1.** *Let $\boldsymbol{\epsilon} \sim \mathcal{N}(\mathbf{0}, \mathbf{I})$ be a sampled noise in a forward process parameterized by $\{\bar{\alpha}_t\}_{t=1}^T$ to get noisy data $\mathbf{x}_t$, the connection between score function $\nabla_{\mathbf{x}_t} \log p(\mathbf{x}_t)$ and noise $\boldsymbol{\epsilon}$ can be given by:*

$$\nabla_{\mathbf{x}_t} \log p(\mathbf{x}_t) = -\frac{\boldsymbol{\epsilon}}{\sqrt{1 - \bar{\alpha}_t}}. \tag{6}$$

Recall that our Proposition 1 makes no assumptions on the form of data $\mathbf{x}$, thus it holds for the noisy data $\mathbf{x}_t$ at **any single timestep** $t$. Applying Eq. (6) to Eq. (4) at timestep $t$, we get the following:

$$-\frac{\boldsymbol{\epsilon}}{\sqrt{1 - \bar{\alpha}_t}} = \sum_y p(y \mid \mathbf{x}_t)\nabla_{\mathbf{x}_t} \log p(\mathbf{x}_t \mid y). \tag{7}$$

For diffusion models, the conditional score function $\nabla_{\mathbf{x}_t} \log p(\mathbf{x}_t \mid y)$ can be estimated as follows:

$$\nabla_{\mathbf{x}_t} \log p(\mathbf{x}_t \mid y) \approx -\frac{\boldsymbol{\epsilon}_\phi(\mathbf{x}_t, t, \mathbf{c}_y)}{\sqrt{1 - \bar{\alpha}_t}}, \tag{8}$$

and therefore we can combine Eq. (7) and Eq. (8) to give a structured estimation in the latent space:

$$\boldsymbol{\epsilon} \approx \sum_y p(y \mid \mathbf{x}_t)\boldsymbol{\epsilon}_\phi(\mathbf{x}_t, t, \mathbf{c}_y), \tag{9}$$

where all score functions are now replaced by noise predictors. Note that $p(y \mid \mathbf{x}_t)$ is the implicit prior of a conditional diffusion model at a single timestep $t$, and thus can be learned by directly plugging the task model prediction on $\mathbf{x}_0$, dubbed as $p_\theta(y \mid \mathbf{x}_0)$, into Eq. (9). The objective to minimize is:

$$\mathcal{L}_{DUSA}(\theta, \phi) = \mathbb{E}_{\boldsymbol{\epsilon}}\Big[\|\boldsymbol{\epsilon} - \sum_y p_\theta(y \mid \mathbf{x}_0)\boldsymbol{\epsilon}_\phi(\mathbf{x}_t, t, \mathbf{c}_y)\|_2^2\Big]. \tag{10}$$

Intuitively, the optimization of this objective encourages the task model to extract knowledge from the semantic structure of a capacious diffusion model, promising better robustness for adaptation.

**Corollary 2.** *The objective in Eq. (10) is extensive to $\mathbf{x}_0$-prediction or $\mathbf{v}$-prediction [50] in diffusion.*

With a total of $K$ classes and $T$ timesteps and computational burden primarily borne by diffusion models, our DUSA shows a task-relevant time complexity of $\mathcal{O}(K)$, while that of Diffusion-TTA is the diffusion-relevant $\mathcal{O}(T)$. Enhanced by practical designs in Sec. 3.3, we empirically find that our DUSA establishes leading performance even with a small budget, leaving Diffusion-TTA behind.

**Free lunch in modern diffusion models.** The proposed objective in Eq. (10) requires the joint training of task model $f_\theta(\mathbf{x})$ and diffusion model $\epsilon_\phi(\mathbf{x}_t, t, \mathbf{c}_y)$ over all conditions $\{\mathbf{c}_y : y \in \mathcal{Y}\}$ simultaneously. The training inefficiency largely stems from the excessive adaptation of the diffusion model, which may not always require knowledge from the task model.

Indeed, Eq. (10) can be interpreted from two distinct perspectives. From one viewpoint, the task model extracts knowledge from the implicit priors of the diffusion model. From another, a weighted optimization is applied to conditional noise estimations, allowing the diffusion model to adapt to the incoming test-time data based on task model predictions. Alternatively, the adaptation of diffusion models can be achieved by introducing unconditional noise estimations with null condition $\varnothing$:

$$\epsilon \approx \epsilon_\phi(\mathbf{x}_t, t, \varnothing). \tag{11}$$

Combining Eq. (9) and Eq. (11) reveals another semantic structure within the diffusion model:

$$\epsilon_\phi(\mathbf{x}_t, t, \varnothing) \approx \sum_y p(y \mid \mathbf{x}_t) \epsilon_\phi(\mathbf{x}_t, t, \mathbf{c}_y), \tag{12}$$

where $\epsilon_\phi(\mathbf{x}_t, t, \varnothing)$ and $\{\epsilon_\phi(\mathbf{x}_t, t, \mathbf{c}_y) : y \in \mathcal{Y}\}$ represent noise estimations from a specific diffusion model capable of handling both unconditional and conditional generation. Note that the implicit priors $\{p(y \mid \mathbf{x}_t) : y \in \mathcal{Y}\}$ in Eq. (12) serve as a critical link between the unconditional and conditional noise estimations. Therefore, an unconditional adaptation of the diffusion model implicitly facilitates its conditional adaptation to the test-time scenario, without reliance on the task model.

Modern conditional diffusion models [20, 22] maintain their unconditional generation capability by employing an additional null condition that replaces the original class conditions with a certain probability (typically $10\%$). Leveraging this feature, we can adjust the objective to enhance efficiency:

$$\mathcal{L}_{cond}(\theta) = \mathbb{E}_\epsilon\left[\|\epsilon - \sum_y p_\theta(y \mid \mathbf{x}_0)\epsilon_\phi(\mathbf{x}_t, t, \mathbf{c}_y)\|_2^2\right], \quad \mathcal{L}_{uncond}(\phi) = \mathbb{E}_\epsilon\left[\|\epsilon - \epsilon_\phi(\mathbf{x}_t, t, \varnothing)\|_2^2\right],$$

$$\mathcal{L}_{DUSA\text{-}U}(\theta, \phi) = \mathcal{L}_{cond}(\theta) + \mathcal{L}_{uncond}(\phi), \tag{13}$$

where the diffusion model is now unconditionally adapted. An immediate concern is whether such modifications would impact adaptation performance. We empirically find it can significantly boost training efficiency with minor to no performance degradation, please refer to Sec. 4.1 for more details.

**Readily applicable to dense prediction tasks.** It is worth noting that our method is not confined to classification tasks, but can be easily applied to a handful of dense prediction tasks as well. Taking semantic segmentation as an example, the task model $p_\theta(\mathbf{y} \mid \mathbf{x}_0)$ is now a dense labeler assigning per-pixel class labels to the input, where $\mathbf{y}$ is the predicted segmentation map of shape $H \times W \times K$, $H \times W$ is the size of input image $\mathbf{x}_0$, and $K$ is the number of interested classes. Again, our proposition is nowhere strict on the form of data, and therefore should be readily applicable to every single pixel in $\mathbf{x}_0$. The new objective to minimize is then easily obtained by utilizing Eq. (4) in a per-pixel fashion:

$$\mathcal{L}_{DUSA\text{-}seg}(\theta, \phi) = \mathbb{E}_{\epsilon, (h,w)}\left[\|\epsilon - \sum_{k=1}^K p_\theta(\mathbf{y} \mid \mathbf{x}_0)_{h,w,k} \cdot \epsilon_\phi(\mathbf{x}_t, t, \mathbf{c}_k)_{h,w}\|_2^2\right], \tag{14}$$

where $(h, w)$ denotes the pixel location in an image sample of size $H \times W$, and $\mathbf{c}_k$ represents the class embedding of a class $k$ in the segmentation task. We highlight that per-pixel noise can be efficiently acquired by extracting elements from the image-level noise estimation $\epsilon_\phi(\mathbf{x}_t, t, \mathbf{c}_k)$, which takes the entire data sample $\mathbf{x}_t$ and class-wise condition $\mathbf{c}_k$ as inputs. As a vast majority of diffusion models are trained with image-level annotations, this design is advantageous as it allows the use of off-the-shelf diffusion models without modifying their training schemes. In contrast, Diffusion-TTA [32] requires the integration of per-pixel conditions into diffusion models to accommodate dense prediction tasks.

### 3.3 Improving Adaptation Efficiency with Practical Designs

**Identifying appropriate timestep.** Since our DUSA intends to extract structured semantic priors from a single timestep, a critical question emerges: which timestep should we utilize to maximize adaptation performance? Iterating over all $T$ timesteps for a certain task model on a specific task is just not practical, and thus a universal preference must be advocated. While the semantic structure uncovered in Eq. (4) is valid for all timesteps in theory, the estimation of score functions by denoising diffusion models [20, 22] can be unreliable. As pointed out by [51, 52], for diffusion models we have

a scheduled $\bar{\alpha}_t$ decreasing with $t$ and $\bar{\alpha}_t \to 1$ when $t \to 0$, which directly amplifies error in score estimation by Eq. (9) at smaller timesteps. A large timestep is also not recommended, as denoising at higher noise levels is more challenging [38], posing a greater challenge to score estimation. Based on the preceding conclusions, we select timestep $t = 100$ and find it suits well for all our experiments.

**Utilizing task model for candidate selection.** Similar to Diffusion Classifiers [25, 29], our DUSA has a computational complexity that scales proportional to class number. To circumvent the slowdown by a large number of classes, we utilize task model prediction to significantly improve adaptation efficiency. With the observation that classifiers typically maintain a $top$-$k$ accuracy [53] and $top$-1 accuracy is our main concern in the test-time adaptation of discriminative models, we opt to apply our DUSA to the most promising classes. Specifically, we deem the posterior $p(y \mid \mathbf{x})$ of less likely class candidates to be zero and only optimize the semantic structure among the selected classes.

A mere decrease in class number can lead to optimization issues, as the task model can be biased towards certain classes, especially with a small batch size. This can be blamed on the underutilization of the semantic structure, further exacerbated by the erratic task model prediction on the pruned classes due to a lack of constraints. To tackle this, we devise a Candidate Selection Module (CSM). In the module, we first adopt LogitNorm [54, 55] to force constraints on the pruned classes to stabilize training, where the logits output of the task model are $\ell_2$ normalized before selection. Intuitively, we discourage the optimization in the magnitude of logits to mitigate overconfidence, especially on pruned classes. Then we handle the class bias problem by introducing randomness in selection. In detail, with a selection budget $b = k + m$, we split it into two parts: $k$ for $top$-$k$ classes select, and $m$ for a multinomial selection without replacement from the remaining classes, where the sampling probabilities are calculated from the logits before normalization. We only focus on the selected $b$ normalized logits, and apply softmax to get their probabilities for our DUSA objective. After a series of practical designs, we succeed in reducing the time complexity of DUSA from $\mathcal{O}(K)$ to $\mathcal{O}(b)$, where a small $b$ should be valid for a large number of classes, as will be shown in Sec. 4.4.

## 4 Experiments

**Datasets and models.** Our experiments are conducted on three benchmarks: ImageNet-C [31] for fully and continual test-time classification, ADE20K [56] with corruptions defined in [31] (dubbed as ADE20K-C) for test-time semantic segmentation. All image corruptions are at the highest severity level 5. We use ResNet-50 [5], ViT-B/16 [57], ConvNeXt-L [6] pre-trained on ImageNet for ImageNet-C experiments, and SegFormer-B5 [58] pre-trained on ADE20K for ADE20K-C ones. We follow [59] and use the GN variant of ResNet-50 for stability. More details are in Appendix F.1.

**Compared methods.** We compare our DUSA with Tent [44], CoTTA [45], EATA [60], SAR [59], RoTTA [61], and Diffusion-TTA [32] for image classification. For semantic segmentation, we compare with BN Adapt [62, 63], Tent [44] and CoTTA [45]. More details are in Appendix F.2.

**Evaluation metrics.** $Top$-1 accuracy (Acc) is reported on each corruption type for image classification. For semantic segmentation, the mean Intersection-over-Union (mIoU) is reported. The main results of our DUSA all come with mean and standard deviation statistics over 3 independent runs.

**Implementation details.** The batch size is $64$ for test-time classification tasks unless otherwise stated. Following [32], we use Adam optimizer [64] with a learning rate of $0.00001$ for our DUSA and Diffusion-TTA. For other baselines, SGD with momentum $0.9$ or Adam optimizer is used in line with the literature [44, 59, 60]. As for test-time semantic segmentation, the batch size is $1$ and Adam with a learning rate of $0.00006/8$ is used, following [45]. We use ImageNet [65] trained DiT [22] and ADE20K trained ControlNet [48] as diffusion models. More details are in Appendix F.3.

### 4.1 Fully Test-Time Adaptation of ImageNet Pre-trained Classifiers

Table 1 shows our DUSA in comparison with relevant methods under the online setting of test-time adaptation to every single corruption domain in ImageNet-C, also known as fully TTA [44]. Generally, our DUSA and Diffusion-TTA both achieve a substantial performance gain, thanks to the knowledge from a capacious generative diffusion model. Sepecifically, our DUSA yields a significant improvement of $+21.9\%$, $+23.3\%$, $+15.7\%$ on ResNet-50, ViT-B/16, and ConvNeXt-L over pre-trained classifiers. Besides, DUSA consistently outperforms the multi-timestep enhanced

Table 1: *Fully test-time adaptation* of ImageNet classifiers on ImageNet-C. The best results are in bold and runner-ups are underlined. GN/LN is short for Group/Layer normalization.

| | Noise | | | Blur | | | | Weather | | | | Digital | | | | |
|---|---|---|---|---|---|---|---|---|---|---|---|---|---|---|---|---|
| Method | Gauss. | Shot | Impul. | Defoc. | Glass | Motion | Zoom | Snow | Frost | Fog | Brit. | Contr. | Elastic | Pixel | JPEG | Avg. |
| ResNet-50 (GN) | 22.1 | 23.0 | 22.0 | 19.8 | 11.4 | 21.5 | 25.0 | 40.3 | 47.0 | 34.0 | 68.8 | 36.3 | 18.5 | 29.3 | 52.6 | 31.4 |
| • Tent | 25.3 | 29.1 | 24.5 | 14.9 | 9.9 | 21.6 | 22.3 | 27.5 | 32.1 | 3.5 | 69.9 | 42.0 | 10.3 | 48.6 | 54.6 | 29.1 |
| • CoTTA | 22.1 | 23.0 | 22.0 | 19.8 | 11.4 | 21.5 | 25.1 | 40.3 | 47.0 | 34.0 | 68.8 | 36.4 | 18.5 | 29.3 | 52.6 | 31.5 |
| • EATA | 38.6 | 40.9 | 39.7 | 27.3 | 26.7 | 36.5 | 38.6 | 50.8 | 49.1 | 55.6 | 72.0 | 49.9 | 40.5 | 55.7 | 58.2 | 45.3 |
| • SAR | 39.6 | 42.4 | 41.0 | 19.8 | 22.9 | 37.1 | 38.7 | 27.3 | 47.4 | 55.1 | 72.4 | 48.8 | 7.2 | 54.9 | 57.4 | 40.8 |
| • RoTTA | 22.8 | 23.8 | 22.5 | 19.7 | 12.0 | 21.8 | 25.2 | 41.3 | 47.5 | 34.6 | 69.2 | 36.8 | 19.2 | 29.9 | 52.9 | 31.9 |
| • Diffusion-TTA | 42.0 | 44.6 | 42.4 | **38.3** | **39.5** | 46.9 | 48.2 | 56.5 | **56.3** | 60.0 | 72.6 | 45.6 | **57.9** | 61.4 | 58.0 | 51.3 |
| • DUSA (Ours) | **45.2**±0.0 | **47.3**±0.0 | **46.3**±0.1 | 37.3±0.1 | 37.6±0.2 | **48.4**±0.0 | **50.3**±0.3 | **59.1**±0.1 | 55.6±0.0 | **63.3**±0.3 | **73.3**±0.0 | **55.1**±0.0 | 56.5±0.3 | **63.2**±0.1 | **60.9**±0.2 | **53.3** |
| • DUSA-U (Ours) | 45.0±0.1 | 47.1±0.1 | 46.1±0.0 | 36.8±0.2 | 37.7±0.1 | 47.9±0.1 | 49.5±0.3 | 59.0±0.1 | 55.4±0.1 | 63.0±0.2 | 73.1±0.1 | 54.3±0.0 | 56.4±0.2 | 62.9±0.1 | 60.5±0.1 | 53.0 |
| ViT-B/16 (LN) | 38.3 | 35.4 | 38.1 | 29.5 | 24.2 | 32.8 | 30.5 | 36.4 | 45.0 | 50.4 | 68.3 | 22.5 | 39.4 | 52.7 | 53.5 | 39.8 |
| • Tent | 53.9 | 54.5 | 54.1 | 44.4 | 47.2 | 53.8 | 6.7 | 4.6 | 61.9 | 65.4 | 72.9 | 54.9 | 58.0 | 65.1 | 64.1 | 50.8 |
| • CoTTA | 38.3 | 35.4 | 38.1 | 29.5 | 24.2 | 32.8 | 30.5 | 36.4 | 45.0 | 50.4 | 68.3 | 22.5 | 39.4 | 52.7 | 53.5 | 39.8 |
| • EATA | 55.4 | 56.3 | 55.3 | 48.9 | 53.4 | 58.6 | 58.2 | 63.5 | 64.1 | 67.5 | 74.3 | 56.5 | 65.7 | 68.5 | 66.6 | 60.9 |
| • SAR | 53.9 | 54.3 | 54.1 | 46.0 | 47.8 | 54.2 | 49.4 | 28.2 | 61.4 | 64.3 | 72.8 | 54.3 | 59.2 | 64.8 | 63.5 | 55.2 |
| • RoTTA | 42.6 | 39.9 | 42.9 | 30.6 | 26.4 | 34.8 | 31.7 | 39.2 | 47.8 | 52.4 | 68.8 | 23.3 | 42.0 | 55.0 | 54.0 | 42.1 |
| • Diffusion-TTA | 52.1 | 54.5 | 53.5 | 49.3 | 52.9 | 56.9 | 55.6 | 60.6 | 63.0 | 64.2 | 72.6 | 47.4 | 66.4 | 67.6 | 62.5 | 58.6 |
| • DUSA (Ours) | **56.6**±0.2 | **57.9**±0.2 | **57.0**±0.0 | **53.3**±0.1 | **56.7**±0.3 | **62.4**±0.1 | **61.6**±0.1 | **65.9**±0.1 | **65.7**±0.1 | **70.1**±0.1 | **75.3**±0.1 | **60.2**±0.1 | **67.9**±0.1 | **69.7**±0.1 | **65.8**±0.1 | **63.1** |
| • DUSA-U (Ours) | 56.3±0.1 | 57.6±0.1 | 56.7±0.1 | 52.5±0.1 | 56.4±0.1 | 61.9±0.1 | 60.4±0.2 | 65.8±0.2 | 65.4±0.1 | 70.0±0.1 | 75.3±0.0 | 58.7±0.2 | 67.8±0.1 | 69.4±0.2 | 64.3±0.1 | 62.6 |
| ConvNeXt-L (LN) | 56.7 | 56.2 | 58.3 | 35.1 | 20.7 | 47.6 | 43.5 | 58.9 | 59.8 | 48.0 | 76.6 | 55.7 | 34.0 | 42.3 | 63.3 | 50.5 |
| • Tent | 57.4 | 57.8 | 58.9 | 35.7 | 24.3 | 51.3 | 46.3 | 59.8 | 58.4 | 11.0 | 77.1 | 61.2 | 35.1 | 50.0 | 64.4 | 49.9 |
| • CoTTA | 56.7 | 56.2 | 58.3 | 35.1 | 20.7 | 47.6 | 43.5 | 59.0 | 59.9 | 48.0 | 76.6 | 55.7 | 34.0 | 42.3 | 63.3 | 50.5 |
| • EATA | 57.5 | 58.0 | 59.0 | 38.7 | 27.1 | 51.6 | 47.0 | 60.7 | 58.5 | 49.3 | 77.2 | 61.3 | 40.2 | 50.3 | 64.5 | 53.4 |
| • SAR | 57.0 | 56.7 | 58.8 | 37.4 | 26.6 | 50.9 | 46.3 | 60.1 | 57.6 | 12.4 | 77.0 | 61.9 | 37.1 | 51.4 | 64.1 | 50.4 |
| • RoTTA | 57.0 | 56.7 | 58.7 | 35.1 | 21.3 | 48.0 | 44.0 | 59.5 | 60.0 | 48.9 | 76.6 | 56.8 | 34.6 | 43.1 | 63.4 | 50.9 |
| • Diffusion-TTA | 58.7 | 59.6 | 58.3 | 50.3 | 48.8 | 57.6 | 54.8 | 63.3 | 64.8 | 68.6 | 77.4 | 60.9 | 62.0 | 65.6 | 65.5 | 61.1 |
| • DUSA (Ours) | **64.2**±0.1 | **65.5**±0.1 | **65.6**±0.1 | **54.7**±0.1 | **53.6**±0.2 | **63.8**±0.1 | **61.9**±0.1 | **70.1**±0.1 | **66.6**±0.2 | **72.7**±0.3 | **79.7**±0.0 | **68.9**±0.0 | **66.1**±0.2 | **70.7**±0.2 | **69.3**±0.1 | **66.2** |
| • DUSA-U (Ours) | 63.8±0.1 | 65.2±0.0 | 65.2±0.1 | 54.0±0.1 | 53.3±0.2 | 63.3±0.1 | 60.6±0.1 | 69.9±0.1 | 66.4±0.1 | 72.5±0.2 | 79.6±0.0 | 68.1±0.0 | 65.9±0.2 | 70.3±0.0 | 68.7±0.1 | 65.8 |

Table 2: *Continual test-time adaptation* of ImageNet pre-trained ConvNext-L on ImageNet-C. The best results are in bold and runner-ups are underlined. LN is short for Layer normalization.

| Time | $t$ ⟶ | | | | | | | | | | | | | | | |
|---|---|---|---|---|---|---|---|---|---|---|---|---|---|---|---|---|
| Method | Gauss. | Shot | Impul. | Defoc. | Glass | Motion | Zoom | Snow | Frost | Fog | Brit. | Contr. | Elastic | Pixel | JPEG | Avg. |
| ConvNeXt-L (LN) | 56.7 | 56.2 | 58.3 | 35.1 | 20.7 | 47.6 | 43.5 | 58.9 | 59.8 | 48.0 | 76.6 | 55.7 | 34.0 | 42.3 | 63.3 | 50.5 |
| • Tent | 57.4 | 60.0 | 62.9 | 38.7 | 32.8 | 53.7 | 50.0 | 60.3 | 60.2 | 67.4 | 77.5 | 64.9 | 23.4 | 52.3 | 64.6 | 55.1 |
| • CoTTA | 56.7 | 56.2 | 58.3 | 35.1 | 20.7 | 47.6 | 43.5 | 59.0 | 59.9 | 48.1 | 76.6 | 55.8 | 34.1 | 42.3 | 63.3 | 50.5 |
| • SAR | 57.0 | 59.6 | 62.6 | 40.9 | 32.5 | 55.1 | 51.1 | 61.1 | 61.2 | 68.5 | 78.0 | 65.4 | 28.4 | 52.1 | 65.2 | 55.9 |
| • EATA | 57.6 | 61.0 | 63.5 | 42.5 | 35.2 | 55.3 | 52.4 | 62.3 | 62.9 | 68.6 | 78.3 | 66.1 | 46.2 | 56.7 | 66.9 | 58.3 |
| • RoTTA | 57.0 | 58.2 | 60.9 | 34.2 | 24.5 | 47.9 | 45.3 | 60.9 | 62.5 | 51.7 | 74.9 | 49.8 | 39.3 | 42.6 | 62.5 | 51.5 |
| • Diffusion-TTA | 58.1 | 63.2 | 63.2 | 54.1 | 56.6 | 61.8 | 62.5 | 65.2 | 65.5 | 68.1 | 75.3 | 58.9 | 37.3 | 54.8 | 60.9 | 60.4 |
| • DUSA (Ours) | **64.1**±0.1 | **67.7**±0.0 | **68.3**±0.1 | **54.8**±0.3 | 56.2±0.2 | **64.6**±0.0 | **65.6**±0.1 | **69.8**±0.0 | **69.9**±0.2 | **74.5**±0.1 | **79.0**±0.1 | **70.3**±0.0 | **68.5**±0.1 | **71.9**±0.1 | **70.7**±0.2 | **67.7** |

Diffusion-TTA by $+2.0\%$, $+4.5\%$, $+5.1\%$ on these classifiers, justifying the exploration of semantic priors underneath the diffusion score estimations and demonstrating a clear superiority of our DUSA.

Furthermore, a thrilling finding is that our DUSA is not reliant on the integration of task model prediction in objective Eq. (10) to maintain the powerful semantic priors implicitly embedded. In Table 1, we provide the adaptation results from Eq. (13), namely DUSA-U. Despite the diffusion model being trained unconditionally in DUSA-U and thus having no chance to borrow knowledge from task models, the performance is still on par with DUSA. This reinforces our conviction that diffusion models inherently possess such semantic priors, even without explicit inclusion of task models. Although DUSA-U is more lightweight (diffusion model only trained on null condition), we stick to DUSA when benchmarking against Diffusion-TTA to ensure fairness in the training budget $b$.

### 4.2 Continual Test-Time Adaptation of ImageNet Pre-trained Classifiers

We also experiment under the online continual test-time adaptation protocol, where the task model should adapt to continually changing scenarios [45]. The outcomes are reported in Table 2. Our DUSA withstands a long period of adaptation and outperforms Diffusion-TTA by a large margin of $+7.3\%$ on ConvNeXt-L. During adaptation, Diffusion-TTA witnesses a performance drop when the OOD datastream type shifts from Weather (Brit.) to Digital (Contr.), while our DUSA shows remarkable adaptation stability. This suggests that DUSA effectively learns from robust and transferable semantic priors from score-based generative modeling, allowing it to shine over prolonged adaptation.

### 4.3 Fully Test-Time Adaptation of ADE20K Pre-trained Segmentors

The versatility of our DUSA to dense prediction tasks is evaluated by fully test-time semantic segmentation on the ADE20K dataset with corruptions. We experiment on SegFormer-B5 and report

Table 3: *Test-time semantic segmentation* of ADE20K pre-trained SegFormer-B5 on ADE20K-C. The best results are in bold and runner-ups are underlined. LN/BN is short for Layer/Batch normalization.

| Method | Noise | | | Blur | | | | Weather | | | | Digital | | | | Avg. |
|---|---|---|---|---|---|---|---|---|---|---|---|---|---|---|---|---|
| | Gauss. | Shot | Impul. | Defoc. | Glass | Motion | Zoom | Snow | Frost | Fog | Brit. | Contr. | Elastic | Pixel | JPEG | |
| Segformer-B5 (LN+BN) | 14.2 | 15.8 | 15.6 | 23.1 | 16.8 | 22.5 | 10.3 | 22.3 | 21.5 | 38.6 | 42.0 | 23.1 | 24.5 | 33.1 | 35.3 | 23.9 |
| • BN Adapt | 10.8 | 12.0 | 11.7 | 16.6 | 12.8 | 16.6 | 7.9 | 17.0 | 16.8 | 29.6 | 32.4 | 18.2 | 19.2 | 25.5 | 26.3 | 18.2 |
| • Tent | 11.2 | 13.0 | 12.5 | 17.0 | 13.5 | 16.9 | 7.7 | 17.7 | 17.4 | 29.7 | 32.5 | 18.6 | 20.0 | 25.8 | 26.4 | 18.7 |
| • CoTTA | 14.6 | 16.1 | 15.8 | 22.6 | 16.5 | 22.1 | 9.8 | 20.9 | 20.4 | 38.8 | 42.3 | 21.9 | 24.3 | 33.6 | 35.4 | 23.7 |
| • DUSA (Ours) | **23.6**$_{\pm1.3}$ | **24.5**$_{\pm1.0}$ | **23.2**$_{\pm0.3}$ | **24.7**$_{\pm0.5}$ | **23.2**$_{\pm1.2}$ | **24.7**$_{\pm0.6}$ | **12.5**$_{\pm0.6}$ | **27.3**$_{\pm1.2}$ | **26.7**$_{\pm0.8}$ | **39.3**$_{\pm0.2}$ | **42.6**$_{\pm0.3}$ | **27.1**$_{\pm1.2}$ | **30.6**$_{\pm0.6}$ | **35.7**$_{\pm0.7}$ | **35.6**$_{\pm0.7}$ | **28.1** |

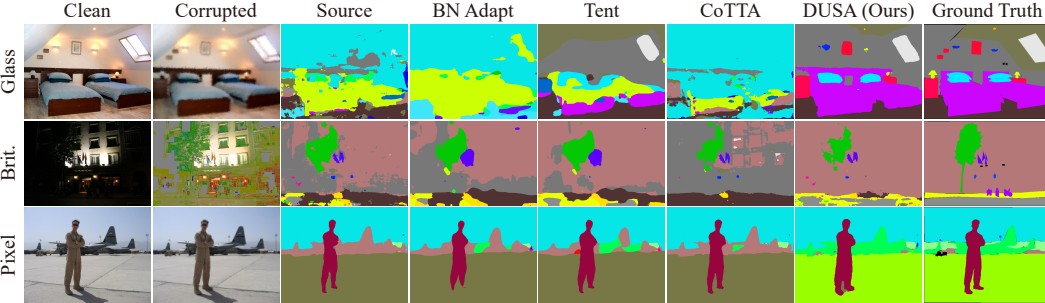

Figure 2: Visualization of segmentation results on ADE20K-C. From left to right: clean and corrupted images, results of the source model, BN Adapt, Tent, CoTTA, our DUSA, and ground-truth labels.

the results in Table 3. Notably, previous methods fail on most tasks, except for CoTTA which achieves modest improvements on a few tasks through a combination of stochastic weight restoration and data augmentation. For all tasks involved, our DUSA takes the lead by exploiting the semantic priors from the high-dimensional latent space of a pre-trained **text-to-image** diffusion model, justifying the extensiveness of our proposition to a wider range of discriminative tasks beyond classification. Segmentation results for all the methods involved are visualized in Fig. 2. Our DUSA, as shown in the illustration, showcases its ability to address errors by incorporating semantic priors from diffusion models, overcoming a limitation faced by other methods that depend solely on the task model's precision. We provide more visualizations over a wide range of scenarios in Appendix I.

### 4.4 Ablation Study

**Selection of timestep** $t$**.** As discussed in Sec. 3.2, our DUSA significantly reduces the number of timesteps to a single timestep of diffusion models. Fig. 3 illustrates the influence of timestep selection on DUSA through adapting the ConvNeXt-L classifier to corruptions from the four main categories. Consistent with our analysis in Sec. 3.3, the guidance from diffusion models is far from perfect when the chosen timestep is either too small ($t \rightarrow 0$) or too large ($t \rightarrow T$). We empirically find that $t = 100$ shows a good performance here, and generalizes well to other backbones and tasks as well. The other timesteps, e.g., $t = 50$, however also emerge as strong contenders and outperform Diffusion-TTA by a considerable margin. For simplicity, we adopt $t = 100$ in all our experiments.

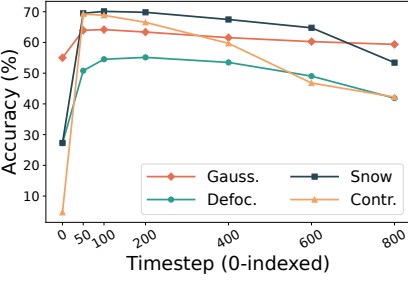

Figure 3: Accuracy of ConvNeXt-L across different selections of timestep.

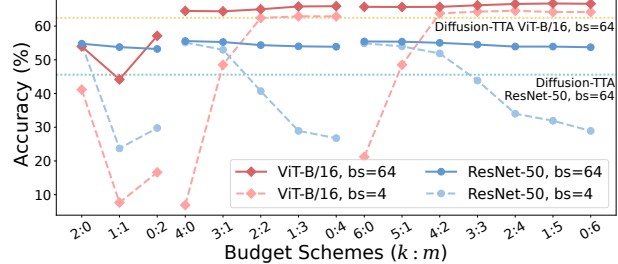

Figure 4: Accuracy of ViT-B/16 on JPEG and ResNet-50 on Contrast, across different budgets for adaptation.

**Effect of components in DUSA.** To grasp a deeper understanding of DUSA, we provide a detailed ablation of critical designs in Table 4, where the number in parentheses means the budget $b = k + m$ allowed for diffusion model forward (D.F.) in Eq. (10), i.e., number of classes to adapt for each sample. The results are obtained on ResNet-50 and ConvNeXt-L over the corruptions within the Noise category. We also include Pixelate corruption results on ConvNeXt-L to offer a well-rounded understanding, upon which the transferability of diffusion models to OOD data is demonstrated.

Before adaptation, the source-only models serve as the baseline. Introducing the objective in Eq. (10) when freezing the diffusion model brings about a performance gain of $+3.6\%$ on ResNet-50 for Noise, but causes a degradation of $-13\%$ on ConvNeXt-L. This is largely due to the instability from discarding classes, as pointed out in Sec. 3.3. Applying LogitNorm instantly mitigates the issue and brings about a consistent gain of $+21.6\%$ and $+3.8\%$ against baselines. The im-

Table 4: Ablation on critical components in DUSA. Components in colored rows are not carried over to subsequent rows.

| | | | | | ResNet-50 | ConvNeXt-L | |
| | | | | | Noise | Noise | Pixel |
| Variants | $k$ | $m$ | D.F. | D.B. | | | |
|---|---|---|---|---|---|---|---|
| Source-only | 0 | 0 | 0 | 0 | 22.4 | 57.1 | 42.3 |
| + score priors inspired loss (4) | 4 | 0 | 4 | 0 | 26.0 | 44.1 | 9.2 |
| + LogitNorm (4) | 4 | 0 | 4 | 0 | 44.0 | 60.9 | 9.6 |
| + adapt diffusion (4) | 4 | 0 | 4 | 4 | 41.2 | 57.8 | 49.3 |
| + LogitNorm (4) | 4 | 0 | 4 | 4 | 46.4 | 65.0 | 70.4 |
| + LogitNorm (6) | 6 | 0 | 6 | 6 | 46.5 | 65.1 | 70.7 |
| + uniform select (6) | 4 | 2 | 6 | 6 | 46.2 | 65.1 | 70.7 |
| + multinomial select (6) (DUSA) | 4 | 2 | 6 | 6 | 46.3 | 65.1 | 70.8 |
| + null conditioning (6) (DUSA-U) | 4 | 2 | 7 | 1 | 46.1 | 64.7 | 70.5 |

provement is made without training diffusion models and therefore can be viewed as exploiting the generative semantic priors formed in diffusion pre-training. However, the outcomes are still below the baseline for Pixelate. We conjecture that such corruption might be OOD even for a diffusion model with strong robustness. The further adaptation of diffusion models removes this concern, pushing all results to a competitive level. We attribute this finding to the fast convergence of generative modeling [1] on unseen data, which is favorable to the online nature of test-time adaptation. Again, the inclusion of LogitNorm yields a significantly boosted accuracy at $46.4\%, 65.0\%$ and $70.4\%$.

To provide a basis for the further ablation of budget schemes, the budget is raised from 4 to 6 with a slight increase in performance. A mild drop in accuracy is witnessed when handing the class bias problem in Sec. 3.3 with a budget $m = 2$ used with uniform sampling, which is then improved by our multinomial selection in the penultimate line. We underline that such a design is indispensable for a small batch size, which is also practical [59]. For better consistency, $k = 4, m = 2$ are universally adopted for DUSA in classification, the ratio between them to be delved into below. Interestingly, with access to a diffusion model capable of unconditional generation, DUSA-U could achieve performance comparable to DUSA using Eq. (13), while significantly reducing the computational cost associated with diffusion model backward (D.B.). We believe this observation can back our claim of the existence of structured semantic priors inherently embedded in diffusion models.

**Specifying a budget scheme.** As a justification for our design of the selection strategies in CSM, we take a thorough investigation into the effects of different budget schemes over varied classifiers (ResNet-50 & ViT-B/16) and batch size (4 & 64) in Fig. 4. At a smaller batch size (bs) of 4 (dashed lines), the budget scheme $k : m$ plays a vital role in performance. Concretely, a large $k$ is favorable to weaker task models (ResNets) for eager adaptation, while a proper $m$ is a must to prevent more powerful ones (ViTs) from overfitting to a subset of classes. When the batch size is increased to the standard 64 (solid lines), our DUSA becomes insensitive to budget schemes, and a consistent gain is observed for both classifiers. DUSA with $b = 4$ even exceeds Diffusion-TTA with $b = 6$, underscoring the advanced efficiency made possible by our proposition and practical designs. For a budget scheme, we find $m = 2$ competitive across varied $b$, and stick to $k = 4, m = 2$ for DUSA.

## 5 Related Work

**Test-time adaptation.** Test-time adaptation [44] focuses on improving source data pre-trained model performance on out-of-distribution target data without label access during inference time. Early works lay more emphasis on adapting the activation statistics of batch normalization (BN) [62, 63, 66, 67]. Test-time training [68, 69, 70, 71] methods manage to adapt through devising a test-time self-supervised objective which is also injected into the pre-training stage, resulting in complicated pipelines and increased computational cost. To lessen dependence on source data and extra loss

injection, fully test-time adaptation [44] is advocated to achieve adaptation with only unlabeled target data. Concretely, previous works are majorly based on entropy minimization objectives: Tent [44] directly minimizes entropy on batched data predictions, MEMO [72] proposes marginal entropy minimization via data augmentation, EATA [60] pursues a sample efficient entropy minimization and anti-forgetting regularization, while SAR [59] advocates sharpness-aware and reliable entropy minimization. Other works delve into extensive distributional shifts in TTA [73], e.g., continual adaptation without forgetting [45, 60], correlative data streams [61] and label shifts [59, 74]. Our DUSA is much different as it is not reliant on error-prone entropy-based objectives and rather extracts knowledge from semantic priors of generative models for better adaptation of the task model.

**Generative models for discriminative tasks.**  The long-standing discussion on the connections between generative and discriminative models [1, 75, 76] has inspired a handful of attempts to integrate the two seemingly disparate paradigms [2, 3, 4, 19, 77, 78, 79, 80]. Specifically, a collection of works showcase the impressive power of generative pre-training followed by supervised fine-tuning [15, 24, 27, 81, 82, 83, 84]. Besides, a few works utilize generative models as zero-shot recognizers [25, 29, 85]. Integrating generative modeling into the task of test-time adaptation is gaining traction. Prior works manage to boost task model performance with a variety of generative techniques, including GANs [86], MAEs [87], energy-based [88] and flow-based [89]. The recent prevalence of diffusion models with extraordinary generation capability stimulates a range of works on adapting them for discriminative tasks. As for TTA, two distinct research directions arise. Appreciating the generative power [90, 91] of diffusion models, a series of works propose to adapt samples in the input space [92, 93, 94]. Another largely under-explored direction is to repurpose the generative objective of diffusion models as a proxy of discriminative ability enhancement, with Diffusion-TTA [32] pioneering in incorporating task predictions into the class condition of denoising objective in an inversion [7, 25, 29, 95] style. Our DUSA belongs to the latter direction but is fundamentally different from [32], in that we delve deeper into the semantic priors of diffusion models from the perspective of score functions and achieve better adaptability and versatility.

**Timestep selection in diffusion models.**  The importance of timestep selection in diffusion models has been widely recognized in the literature. In image editing tasks, diffusion models are observed to exhibit a natural coarse-to-fine pattern during the reverse process [91, 96, 97, 98, 99, 100]. Consequently, the trade-off between realism and fidelity in editing largely stems from the chosen intervals of timesteps. In discriminative tasks, timestep selection is also crucial to the quality of extracted features. An early study [83] elucidated the features within diffusion models, revealing that the most informative ones are derived from smaller timesteps. Subsequent works have reinforced this finding by utilizing a single small timestep across various tasks, including semantic segmentation, referring image segmentation, depth estimation [24], object detection [26], semantic correspondence [101], and one-shot image segmentation [102]. While our DUSA aligns with the existing literature on using a single timestep, it differs by extracting semantic priors rather than focusing on feature extraction.

# 6   Conclusion

In this paper, we introduce DUSA, a competitive test-time adaptation method built on the structured semantic priors underlying diffusion models, which serve as score estimators. A proposition is offered to unveil the semantic structure in these score-based models, upon which a test-time objective is derived to fully exploit the implicit semantic priors. Our approach is also shown to generalize well to modern diffusion models and dense prediction tasks. Additionally, we enhance the adaptation efficiency through a few practical designs. The effectiveness of our DUSA is demonstrated across three challenging benchmarks, where it consistently outperforms competing methods. We hope our method will pave the way for better utilization of generative modeling for discriminative tasks.

## Acknowledgments and Disclosure of Funding

This paper was supported by the National Natural Science Foundation of China (No. 62376026), Beijing Nova Program (No. 20230484296) and KuaiShou.

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

## A   Broader Impacts and Limitations

**Broader Impacts.**   In this work, an approach to incorporate generative diffusion models into the discriminative task of test-time adaptation is introduced. The leverage of knowledge from generative modeling teaches a whole picture of the scenario to the task model, enabling better robustness and adaptability and is thus favorable to some relevant scenarios like medical analysis and quality control of manufacturing in varying environments. However, the adoption of generative models might come with a risk of learning from a biased source of knowledge, leading to improper decisions from the adapted task model. A possible remedy for this is to foster the research into more unbiased or balanced training of generative models.

**Limitations.**   Our work presents an effective way of borrowing semantic priors from score-based diffusion models to benefit the discriminative task model that demands adaptation at test time without labels. We acknowledge that, as with any research endeavor, limitations exist in our work. Firstly, our competitive method involves a diffusion model pre-trained at least on the same set of source data as the task model, which may not be easily accessible for certain scenarios. However, we believe such a dilemma can be mitigated with advances in generative diffusion models, which have demonstrated remarkable progress in both generality and robustness. Our approach achieves significant adaptation efficiency gains against prior diffusion-based TTA methods, but there is still a gap in the time compared to methods that only update the task model. Therefore, it may not meet extreme efficiency demands in scenarios like autonomous driving. Our method, however, yields superior performance and is thus appealing to another set of tasks where trading a slight loss in efficiency for boosted accuracy is tolerable, e.g., non-emergent medical diagnosis. With this said, we highlight that our theoretical findings are not confined to diffusion models but extensive to all score-based models, therefore substituting a more lightweight technique in score estimation for the computationally expensive diffusion models can be a promising avenue. We leave it for future work.

## B   Licenses for Existing Assets

All models and datasets used in our experiments are publicly available, and their licenses are listed below. (D) means data, (M) means model, (C) means code.

- (D) ImageNet-C [31]: CC-BY 4.0
- (D) ADE20K 2016 [56]: BSD 3-clause license
- (M) ResNet-50 [5]: Apache-2.0 license
- (M) ViT-Base/16 [57]: Apache-2.0 license
- (M) ConvNeXt-L [6]: MIT license
- (M) DiT-XL/2 [22]: CC-BY-NC 4.0
- (M) Stable Diffusion v1.5 [20]: The CreativeML OpenRAIL M license
- (M) ControlNet [48]: The CreativeML OpenRAIL M license
- (C) MMPreTrain [103]: Apache-2.0 license
- (C) MMSegmentation [104]: Apache-2.0 license
- (C) Diffusion-TTA [32]: MIT license

## C   Proofs

**Proposition 1.** *Let $p(\mathbf{x})$ and $\{p(\mathbf{x} \mid y) : y \in \mathcal{Y}\}$ be continuously differentiable probability densities, their score functions $\nabla_{\mathbf{x}} \log p(\mathbf{x})$ and $\{\nabla_{\mathbf{x}} \log p(\mathbf{x} \mid y) : y \in \mathcal{Y}\}$, the following equation holds:*

$$\nabla_{\mathbf{x}} \log p(\mathbf{x}) = \sum_y p(y \mid \mathbf{x}) \nabla_{\mathbf{x}} \log p(\mathbf{x} \mid y). \tag{15}$$

*Proof.* The proof starts from the law of total probability. With the assumption of continuously differentiable, we can take derivatives of the probability densities on input data. Note that we always

assume the existence of $\mathbf{x}$, implying that $p(\mathbf{x})$ is non-zero. The complete proof is then as follows:

$$p(\mathbf{x}) = \sum_y p(y)p(\mathbf{x} \mid y) \qquad \text{(law of total probability)}$$

$$\nabla_{\mathbf{x}} p(\mathbf{x}) = \sum_y p(y)\nabla_{\mathbf{x}} p(\mathbf{x} \mid y) \qquad \text{(differentiate both sides on } \mathbf{x})$$

$$p(\mathbf{x})\nabla_{\mathbf{x}}\log p(\mathbf{x}) = \sum_y p(y)p(\mathbf{x} \mid y)\nabla_{\mathbf{x}}\log p(\mathbf{x} \mid y) \qquad \text{(log-derivative trick)} \tag{16}$$

$$p(\mathbf{x})\nabla_{\mathbf{x}}\log p(\mathbf{x}) = \sum_y p(\mathbf{x})p(y \mid \mathbf{x})\nabla_{\mathbf{x}}\log p(\mathbf{x} \mid y) \qquad \text{(Bayes theorem)}$$

$$\nabla_{\mathbf{x}}\log p(\mathbf{x}) = \sum_y p(y \mid \mathbf{x})\nabla_{\mathbf{x}}\log p(\mathbf{x} \mid y) \qquad \text{(eliminate } p(\mathbf{x}))$$

$\blacksquare$

**Corollary 1.** *Let $\boldsymbol{\epsilon} \sim \mathcal{N}(\mathbf{0}, \mathbf{I})$ be a sampled noise in a forward process parameterized by $\{\bar{\alpha}_t\}_{t=1}^{T}$ to get noisy data $\mathbf{x}_t$, the connection between score function $\nabla_{\mathbf{x}_t}\log p(\mathbf{x}_t)$ and noise $\boldsymbol{\epsilon}$ can be given by:*

$$\nabla_{\mathbf{x}_t}\log p(\mathbf{x}_t) = -\frac{\boldsymbol{\epsilon}}{\sqrt{1 - \bar{\alpha}_t}}. \tag{17}$$

*Proof.* According to the definition in Eq. (2), we have

$$q(\mathbf{x}_t \mid \mathbf{x}_0) = \mathcal{N}(\mathbf{x}_t; \sqrt{\bar{\alpha}_t}\mathbf{x}_0, (1 - \bar{\alpha}_t)\mathbf{I}). \tag{18}$$

Tweedie's Formula in Lemma 1 states that, for observed gaussian variable $\mathbf{z} \mid \boldsymbol{\mu_z} \sim \mathcal{N}(\mathbf{z}; \boldsymbol{\mu_z}, \boldsymbol{\Sigma_z})$, we have the estimation

$$\mathbb{E}[\boldsymbol{\mu_z} \mid \mathbf{z}] = \mathbf{z} + \boldsymbol{\Sigma_z}\nabla_{\mathbf{z}}\log p(\mathbf{z}). \tag{19}$$

Applying Tweedie's Formula, we now have

$$\mathbb{E}[\boldsymbol{\mu}_{\mathbf{x}_t} \mid \mathbf{x}_t] = \mathbf{x}_t + \boldsymbol{\Sigma}_{\mathbf{x}_t}\nabla_{\mathbf{x}_t}\log p(\mathbf{x}_t)$$
$$\implies \sqrt{\bar{\alpha}_t}\mathbf{x}_0 = \mathbf{x}_t + (1 - \bar{\alpha}_t)\nabla_{\mathbf{x}_t}\log p(\mathbf{x}_t) \tag{20}$$
$$\implies \sqrt{\bar{\alpha}_t}\mathbf{x}_0 - \mathbf{x}_t = (1 - \bar{\alpha}_t)\nabla_{\mathbf{x}_t}\log p(\mathbf{x}_t).$$

Reuse the reparameterized version of Eq. (2), with $\boldsymbol{\epsilon} \sim \mathcal{N}(\mathbf{0}, \mathbf{I})$,

$$\mathbf{x}_t = \sqrt{\bar{\alpha}_t}\mathbf{x}_0 + \sqrt{1 - \bar{\alpha}_t}\boldsymbol{\epsilon}$$
$$\implies \sqrt{\bar{\alpha}_t}\mathbf{x}_0 - \mathbf{x}_t = -\sqrt{1 - \bar{\alpha}_t}\boldsymbol{\epsilon}. \tag{21}$$

Combining Eq. (20) and Eq. (21), we obtain

$$\nabla_{\mathbf{x}_t}\log p(\mathbf{x}_t) = -\frac{\boldsymbol{\epsilon}}{\sqrt{1 - \bar{\alpha}_t}}. \tag{22}$$

$\blacksquare$

**Corollary 2.** *The objective in Eq. (10) is extensive to $\mathbf{x}_0$-prediction or $\mathbf{v}$-prediction [50] in diffusion.*

*Proof.* We provide the proof for $\mathbf{x}_0$-prediction and $\mathbf{v}$-prediction variants separately.

As a recall, the objective Eq. (10) is built on the estimation given by $\boldsymbol{\epsilon}$-prediction in Eq. (9).

For the sake of clarity, here we re-present the formula but neglect the estimation error:

$$\boldsymbol{\epsilon} = \sum_y p(y \mid \mathbf{x}_t)\boldsymbol{\epsilon}_\phi(\mathbf{x}_t, t, \mathbf{c}_y). \tag{23}$$

*For $\mathbf{x}_0$-prediction*, we employ data predictor $\mathbf{x}_\phi(\mathbf{x}_t, t, \mathbf{c})$ to predict $\mathbf{x}_0$.

As we have the formula $\mathbf{x}_t = \sqrt{\bar{\alpha}_t}\mathbf{x}_0 + \sqrt{1 - \bar{\alpha}_t}\boldsymbol{\epsilon}$, it can be deduced that $\forall \mathbf{c} \in \{\mathbf{c}_y : y \in \mathcal{Y}\} \cup \{\varnothing\}$,

$$\mathbf{x}_t = \sqrt{\bar{\alpha}_t}\mathbf{x}_\phi(\mathbf{x}_t, t, \mathbf{c}) + \sqrt{1 - \bar{\alpha}_t}\boldsymbol{\epsilon}_\phi(\mathbf{x}_t, t, \mathbf{c}). \tag{24}$$

Thus we have

$$
\begin{aligned}
\mathbf{x}_0 &= \frac{\mathbf{x}_t - \sqrt{1-\bar{\alpha}_t}\boldsymbol{\epsilon}}{\sqrt{\bar{\alpha}_t}} \\
&= \frac{\mathbf{x}_t - \sqrt{1-\bar{\alpha}_t}\sum_y p(y \mid \mathbf{x}_t)\boldsymbol{\epsilon}_\phi(\mathbf{x}_t, t, \mathbf{c}_y)}{\sqrt{\bar{\alpha}_t}} \quad \text{(Eq. (23))} \\
&= \frac{\sum_y p(y \mid \mathbf{x}_t)\big(\mathbf{x}_t - \sqrt{1-\bar{\alpha}_t}\boldsymbol{\epsilon}_\phi(\mathbf{x}_t, t, \mathbf{c}_y)\big)}{\sqrt{\bar{\alpha}_t}} \quad \big(\sum_y p(y \mid \mathbf{x}_t) = 1\big) \\
&= \frac{\sum_y p(y \mid \mathbf{x}_t)\sqrt{\bar{\alpha}_t}\mathbf{x}_\phi(\mathbf{x}_t, t, \mathbf{c}_y)}{\sqrt{\bar{\alpha}_t}} \quad \text{(Eq. (24))} \\
&= \sum_y p(y \mid \mathbf{x}_t)\mathbf{x}_\phi(\mathbf{x}_t, t, \mathbf{c}_y).
\end{aligned}
\tag{25}
$$

Therefore objective in Eq. (10) can be rewritten as

$$
\mathcal{L}_{DUSA}(\theta, \phi) = \mathbb{E}_{\boldsymbol{\epsilon}}\Big[\|\mathbf{x}_0 - \sum_y p_\theta(y \mid \mathbf{x}_0)\mathbf{x}_\phi(\mathbf{x}_t, t, \mathbf{c}_y)\|_2^2\Big].
\tag{26}
$$

*For* $\mathbf{v}$-*prediction*, we employ velocity predictor $\mathbf{v}_\phi(\mathbf{x}_t, t, \mathbf{c})$ to predict a constructed velocity objective

$$
\mathbf{v}_t = \sqrt{\bar{\alpha}_t}\boldsymbol{\epsilon} - \sqrt{1-\bar{\alpha}_t}\mathbf{x}_0.
\tag{27}
$$

Similar to the treatment above in Eq. (24), we have $\forall \mathbf{c} \in \{\mathbf{c}_y : y \in \mathcal{Y}\} \cup \{\varnothing\}$,

$$
\mathbf{v}_\phi(\mathbf{x}_t, t, \mathbf{c}) = \sqrt{\bar{\alpha}_t}\boldsymbol{\epsilon}_\phi(\mathbf{x}_t, t, \mathbf{c}) - \sqrt{1-\bar{\alpha}_t}\mathbf{x}_\phi(\mathbf{x}_t, t, \mathbf{c}).
\tag{28}
$$

Applying the conclusions in Eq. (23), Eq. (25) and Eq. (28), we have

$$
\begin{aligned}
\mathbf{v}_t &= \sqrt{\bar{\alpha}_t}\boldsymbol{\epsilon} - \sqrt{1-\bar{\alpha}_t}\mathbf{x}_0 \\
&= \sqrt{\bar{\alpha}_t}\sum_y p(y \mid \mathbf{x}_t)\boldsymbol{\epsilon}_\phi(\mathbf{x}_t, t, \mathbf{c}_y) - \sqrt{1-\bar{\alpha}_t}\sum_y p(y \mid \mathbf{x}_t)\mathbf{x}_\phi(\mathbf{x}_t, t, \mathbf{c}_y) \\
&= \sum_y p(y \mid \mathbf{x}_t)\big(\sqrt{\bar{\alpha}_t}\boldsymbol{\epsilon}_\phi(\mathbf{x}_t, t, \mathbf{c}_y) - \sqrt{1-\bar{\alpha}_t}\mathbf{x}_\phi(\mathbf{x}_t, t, \mathbf{c}_y)\big) \\
&= \sum_y p(y \mid \mathbf{x}_t)\mathbf{v}_\phi(\mathbf{x}_t, t, \mathbf{c}_y).
\end{aligned}
\tag{29}
$$

Therefore, objective in Eq. (10) can be rewritten as

$$
\mathcal{L}_{DUSA}(\theta, \phi) = \mathbb{E}_{\boldsymbol{\epsilon}}\Big[\|\mathbf{v}_t - \sum_y p_\theta(y \mid \mathbf{x}_0)\mathbf{v}_\phi(\mathbf{x}_t, t, \mathbf{c}_y)\|_2^2\Big].
\tag{30}
$$

∎

# D   On the Estimation Bias and Efficiency of Diffusion-TTA and DUSA

## D.1   Diffusion-TTA

Prior works [25, 29, 30, 32] rely on the Monte Carlo method for a good estimation of the likelihood.

The core of diffusion model formulation is the evidence lower bound (ELBO), also called variational lower bound (VLB), where the log-likelihood of data is bounded:

$$
\log p_\phi(\mathbf{x}_0 \mid \mathbf{c}) \geq \mathbb{E}_q\Big[\log \frac{p_\phi(\mathbf{x}_{0:T} \mid \mathbf{c})}{q(\mathbf{x}_{1:T} \mid \mathbf{x}_0)}\Big],
\tag{31}
$$

where $q$ is the forward (diffusion) process, and $p_\phi(\cdot)$ is the reverse process modeled by diffusion models parameterized by $\phi$.

The ELBO can be further deduced and simplified with diffusion loss, we have

$$
\begin{aligned}
& \mathbb{E}_q \left[ \log \frac{p_\phi(\mathbf{x}_{0:T} \mid \mathbf{c})}{q(\mathbf{x}_{1:T} \mid \mathbf{x}_0)} \right] \\
& = \mathbb{E}_q \log p_\phi(\mathbf{x}_0 \mid \mathbf{x}_1, \mathbf{c}) - \sum_{t=2}^{T} \mathbb{E}_q D_{KL}(q(\mathbf{x}_{t-1} \mid \mathbf{x}_t, \mathbf{x}_0) \parallel p_\phi(\mathbf{x}_{t-1} \mid \mathbf{x}_t, \mathbf{c})) \\
& \quad - D_{KL}(q(\mathbf{x}_T \mid \mathbf{x}_0) \parallel p_\phi(\mathbf{x}_T)) \\
& = -\mathbb{E}_{\boldsymbol{\epsilon},t} \left[ \sum_{t=2}^{T} w_t \| \boldsymbol{\epsilon} - \boldsymbol{\epsilon}_\phi(\mathbf{x}_t, t, \mathbf{c}) \|_2^2 - \log p_\phi(\mathbf{x}_0 \mid \mathbf{x}_1, \mathbf{c}) \right] + C \\
& \approx -T \mathbb{E}_{\boldsymbol{\epsilon},t} \left[ \| \boldsymbol{\epsilon} - \boldsymbol{\epsilon}_\phi(\mathbf{x}_t, t, \mathbf{c}) \|_2^2 \right] + C. \qquad (\log p_\phi(\mathbf{x}_0 \mid \mathbf{x}_1, \mathbf{c}) \text{ typically small, } w_t \leftarrow 1)
\end{aligned}
\tag{32}
$$

In the equations above, a theoretical approximation is made by simplifying weights $w_t$ to 1 and ignoring the $\log p_\phi(\mathbf{x}_0 \mid \mathbf{x}_1, \mathbf{c})$ term. This simplification introduces **theoretical bias** relative to the true likelihood, as the approximation in Eq. (32) prevents the equality in Eq. (31) from holding.

Therefore, with a uniform prior on label space, the objective of Diffusion-TTA [32] is

$$
\begin{aligned}
\max_y p(y \mid \mathbf{x}) &= \max_y p(\mathbf{x} \mid y) p(y) && \text{(Bayes theorem)} \\
&= \max_y p(\mathbf{x} \mid y) && \text{(uniform prior)} \\
&\gtrapprox \max_y \exp\left( -T \mathbb{E}_{t,\boldsymbol{\epsilon}} \left[ \| \boldsymbol{\epsilon} - \boldsymbol{\epsilon}_\phi(\mathbf{x}_t, t, \mathbf{c}_y) \|_2^2 \right] + C \right) && \text{(Eq. (32))} \\
&= \min_y \mathbb{E}_{t,\boldsymbol{\epsilon}} \left[ \| \boldsymbol{\epsilon} - \boldsymbol{\epsilon}_\phi(\mathbf{x}_t, t, \mathbf{c}_y) \|_2^2 \right].
\end{aligned}
\tag{33}
$$

The objective thus takes expectation over $\boldsymbol{\epsilon}$ and $t$, where Diffusion-TTA finds $t$ plays a crucial rule in the final result and applies Monte Carlo method on 180 samples of $t$, for every single data sample.

### D.2  DUSA

In contrast, our DUSA approach does not rely on the ELBO for likelihood maximization. Instead, we leverage the structured semantic priors introduced in Eq. (4) to guide the task model in extracting knowledge from the diffusion model using a single timestep. Notably, the only estimations in our method are the noise predictions $\boldsymbol{\epsilon} \approx \boldsymbol{\epsilon}_\phi(\mathbf{x}_t, t, \mathbf{c}_y)$ in Eq. (8), which are provably unbiased.

To elaborate, the diffusion models are typically optimized with the following simplified objective:

$$
\mathcal{L}_{simple}(\phi) = \mathbb{E}_{\mathbf{x}_t \mid \boldsymbol{\epsilon}, t} \left[ \| \boldsymbol{\epsilon} - \boldsymbol{\epsilon}_\phi(\mathbf{x}_t, t, \mathbf{c}_y) \|_2^2 \right].
\tag{34}
$$

At its optimal point, the noise estimations should satisfy:

$$
\frac{\partial \mathcal{L}_{simple}(\phi)}{\partial \boldsymbol{\epsilon}_\phi} = \mathbb{E}_{\mathbf{x}_t \mid \boldsymbol{\epsilon}, t}[2(\boldsymbol{\epsilon} - \boldsymbol{\epsilon}_\phi(\mathbf{x}_t, t, \mathbf{c}_y))] = 0,
\tag{35}
$$

which implies the following:

$$
\mathbb{E}_{\mathbf{x}_t \mid \boldsymbol{\epsilon}, t}[\boldsymbol{\epsilon}_\phi(\mathbf{x}_t, t, \mathbf{c}_y)] = \mathbb{E}_{\mathbf{x}_t \mid \boldsymbol{\epsilon}, t}[\boldsymbol{\epsilon}] = \boldsymbol{\epsilon}.
\tag{36}
$$

Therefore, the conditional noise estimation $\boldsymbol{\epsilon}_\phi$ is an **unbiased estimator** of the true noise $\boldsymbol{\epsilon}$:

$$
\text{Bias}(\boldsymbol{\epsilon}_\phi(\mathbf{x}_t, t, \mathbf{c}_y), \boldsymbol{\epsilon}) = \mathbb{E}_{\mathbf{x}_t \mid \boldsymbol{\epsilon}, t}[\boldsymbol{\epsilon}_\phi(\mathbf{x}_t, t, \mathbf{c}_y)] - \boldsymbol{\epsilon} = 0.
\tag{37}
$$

Similiarly, the unconditional noise estimation $\boldsymbol{\epsilon} \approx \boldsymbol{\epsilon}_\phi(\mathbf{x}_t, t, \varnothing)$ in Eq. (11) is also provably unbiased.

# E  Algorithm of DUSA

---

**Algorithm 1:** **D**iff**U**sion **S**core for test-time **A**daptation (DUSA)

---

1: **Input:** Test samples $\{\mathbf{x}_0^j\}_{j=1}^M$, discriminative task model $f_\theta$, generative diffusion model $\boldsymbol{\epsilon}_\phi$, timestep $t$, learning rate $\eta$.
2: **Output:** Task predictions $\{\hat{y}_j\}_{j=1}^M$.
3: **for** $j = 1$ to $M$ **do**
4:     Predict logits $\mathbf{z} = f_\theta(\mathbf{x}_0^j)$ with task model.
5:     Make task prediction $\hat{y}_j = \mathrm{argmax}_y\, \mathbf{z}_y$.
6:     Apply CSM to prune logits $\mathbf{z}$ to $K$ classes and get $\hat{\mathbf{z}}$.
7:     Take softmax over $\hat{\mathbf{z}}$ to get probabilities $\{p_\theta(y \mid \mathbf{x}_0^j)\}_{y=1}^K$.
8:     Compute conditions $\{\mathbf{c}_y\}_{y=1}^K$ for all classes in $\hat{\mathbf{z}}$.
9:     Sample noise $\boldsymbol{\epsilon}$ and get noisy sample $\mathbf{x}_t^j$ as in Eq. (2).
10:     Get conditional noise predictions $\{\boldsymbol{\epsilon}_\phi(\mathbf{x}_t^j, t, \mathbf{c}_y)\}_{y=1}^K$.
11:     Calculate the objective $\mathcal{L}(\theta, \phi)$ in Eq. (10) or Eq. (13).
12:     Update task model weights $\theta \leftarrow \theta - \eta\nabla_\theta\mathcal{L}(\theta, \phi)$.
13:     Update diffusion model weights $\phi \leftarrow \phi - \eta\nabla_\phi\mathcal{L}(\theta, \phi)$.
14: **end for**

---

# F  More Experimental Details

## F.1  More Details on Datasets

**ImageNet-C [31].**  ImageNet-C consists of corrupted images computed from applying algorithmic corruptions to the ImageNet [65] validation set, which has 50,000 images. To construct ImageNet-C, 15 corruptions that fall into 4 categories are applied separately to the whole validation set, including Gaussian noise (Gauss.), shot noise (Shot), impulse noise (Impl.), defocus blur (Defoc.), glass blur (Glass), motion blur (Motion), zoom blur (Zoom), snow (Snow), frost (Frost), fog (Fog), brightness (Brit.), contrast (Contr.), elastic transformation (Elastic), pixelation (Pixel) and JPEG compression (JPEG). Each corruption type has 5 severity levels and a higher severity level means a more severe distribution shift. We use the highest severity level 5 in all our experiments.

**ADE20K [56].**  ADE20K is a semantic segmentation dataset containing more than 20k images annotated with pixel-level labels on instances and object parts, among which 2k images are for validation. A total of 150 semantic classes are benchmarked for evaluation. We apply the corruptions defined in [31] with tools provided by [105] to construct ADE20K-C, which shares the corruption types with ImageNet-C, at the highest severity level 5. We use this corrupted benchmark for test-time semantic segmentation tasks.

## F.2  More Details on Compared Methods

**BN Adapt [62, 63, 66].**  BN Adapt has the straightforward idea that the statistics in Batch Normalization layers are data-dependent, and therefore can be adapted at test time for better generalization.

**Tent [44].**  Tent is a pioneer in addressing the fully test-time adaptation problem. In the work, test-time batch normalization is employed to recalibrate the statistics within Batch Normalization, where features are normalized based on the current batch's statistics. Additionally, Tent utilizes entropy minimization to adjust the affine parameters of Batch Normalization layers.

**CoTTA [45].**  CoTTA focuses on performing test-time adaptation for continually changing distributions. Firstly, it generates more robust and reliable pseudo labels through multiple data augmentations and employs a mean-teacher architecture to reduce error accumulation. Secondly, to prevent catastrophic forgetting, a stochastic restoration strategy is proposed, which randomly rolls back a portion of the student model's parameters. Meanwhile, in the mean-teacher architecture, all the parameters of the student model remain trainable.

**EATA [60].** EATA is devoted to improving efficiency and preventing knowledge forgetting. Concretely, the unreliable and redundant test samples are filtered out by the value of prediction entropy and similarity to the mean prediction. Besides, an anti-forgetting regularization based on fisher importance is proposed. Additionally, only affine parameters in batch normalization layers are adapted.

**SAR [59].** SAR points out the unreliability of entropy minimization and the instability of Batch Normalization layers during test-time adaptation. Motivated by this, it proposes a sharpness-aware and reliable entropy minimization for Group/Layer Normalization-based models. For time efficiency, only affine parameters in Group/Layer Normalization layers are optimized.

**RoTTA [61].** RoTTA is dedicated to conducting test-time adaptation on complex test streams characterized by continually changing data distributions and temporally correlated label distributions. To begin with, it develops a prediction-balanced sampling strategy grounded in uncertainty and timeliness, ensuring the maintenance of a robust snapshot of the test distribution for adaptation. Furthermore, a robust batch normalization layer is devised to recalibrate normalization statistics by applying exponential moving averages to selected samples. Lastly, it introduces a timeliness reweighting strategy to attain stable and robust adaptation. Only affine parameters in robust batch normalization layers undergo training.

**Diffusion-TTA [32].** Diffusion-TTA is proposed to adapt pre-trained task models using feedback from pre-trained diffusion models. The integration of the task model into the diffusion model is achieved by modulating the conditioning of the diffusion model using the output of the task model. With the generative objective of diffusion models, the knowledge is backpropagated through conditioning to the task model. Hundreds of timesteps are sampled in a diffusion model for a single sample as an approximation of the likelihood estimation to improve adaptation performance.

### F.3 More Details on Implementation

All pre-trained models involved in our paper are publicly available, including ResNet-50 (GN)[2], ViT-B/16 (LN)[3], ConvNext-L (LN)[4], DiT-XL/2[5] and ControlNet[6] based on Stable Diffusion v1.5[7] from `timm` [106] or their official repository. The classifiers and DiT-XL/2 are pre-trained on ImageNet, while ADE20K is used to pre-train SegFormer-B5 and ControlNet.

As for code, we (re)implement all test-time adaptation methods for classification under a framework modified from MMPreTrain [103], except for Diffusion-TTA we adopt its official implementation. For test-time semantic segmentation tasks, we (re)implement all methods under a framework modified from MMSegmentation [104].

All experiments performed are with a batch size of $64$, except for part of the analysis in Fig. 4. Our DUSA is trained with a batch size of $8$ and a gradient accumulation of $8$ steps, to yield an effective batch size of $64$. Limited by its implementations, Diffusion-TTA is run with a batch size of $1$, and a gradient accumulation of $64$ steps is applied, also crafting an effective batch size of $64$. We follow [32] and use Adam [64] optimizer with a learning rate of $0.00001$ ($1.0 \times 10^{-5}$) and a weight decay of $0.0$ for both our DUSA and Diffusion-TTA, which applies to all classifiers. As for other compared methods, we use Stochastic Gradient Descent (SGD) with momentum $0.9$ and a learning rate of $0.00025$ ($2.5 \times 10^{-4}$) for ResNet-50, while a learning rate of $0.001$ ($1.0 \times 10^{-3}$) is used for ViT-B/16, in accordance with the literature [44, 59, 60] to obtain decent baseline results. For ConvNext-L, we use Adam with a learning rate of $0.00001$ ($1.0 \times 10^{-5}$) with a weight decay of $0.0$ for all methods involved. The diffusion model for classification tasks is a Diffusion Transformer

---

[2]`https://github.com/rwightman/pytorch-image-models/releases/download/v0.1-rsb-weights/resnet50_gn_a1h2-8fe6c4d0.pth`

[3]`https://storage.googleapis.com/vit_models/augreg/B_16-i1k-300ep-lr_0.001-aug_strong2-wd_0.1-do_0.1-sd_0.1--imagenet2012-steps_20k-lr_0.01-res_224.npz`

[4]`https://dl.fbaipublicfiles.com/convnext/convnext_large_1k_224_ema.pth`

[5]`https://dl.fbaipublicfiles.com/DiT/models/DiT-XL-2-256x256.pt`

[6]`https://huggingface.co/lllyasviel/control_v11p_sd15_seg/blob/main/diffusion_pytorch_model.bin`

[7]`https://huggingface.co/runwayml/stable-diffusion-v1-5/blob/main/v1-5-pruned.ckpt`

DiT-XL/2 [22] trained on ImageNet [65] from scratch. For all classification tasks on ImageNet-C, we adopt the standard pipeline [31] and center crop images to $224 \times 224$ for task models. Our DUSA and Diffusion-TTA both use DiT-XL/2 with an input size of $256 \times 256$ as the diffusion model, therefore we resize the cropped $224 \times 224$ image to $256 \times 256$ before passing it to DiT-XL/2 as input. During adaptation, we freeze the VAE encoder and condition embedder of DiT, while training the denoising transformer which functions in a latent space of $32 \times 32 \times 4$. All parameters of the task models are adapted in DUSA, as is done in Diffusion-TTA and CoTTA.

For hyperparameters in DUSA, we set $t = 100, k = 4$ and $m = 2$ for all classification tasks. For the sake of fair comparison and practical considerations in compute resources, we give Diffusion-TTA the same budget $b = 6$, i.e., 6 timesteps in diffusion models are randomly sampled for the training of each image sample, which applies to all results reported on Diffusion-TTA, including those in Fig. 3 and Fig. 4. For both our DUSA and Diffusion-TTA that involve diffusion models, the noise $\epsilon$ added to input data is randomly sampled. As for other compared methods, we follow all hyperparameters in their original setup and please refer to their paper for more detailed hyperparameter settings.

For test-time semantic segmentation, we follow [45] and use a batch size of 1, Adam optimizer with a learning rate of $0.00006/8$ ($6.0 \times 10^{-5}/8$) and a weight decay of $0.0$ for all methods. The diffusion model for this task is a ControlNet [48] finetuned from Stable Diffusion v1.5 [20] on ADE20K. Note that ControlNets come with extra conditioning capability, but we **dismiss extra conditions beyond text** so that it can be recognized as a typical text-to-image diffusion model. In detail, the conditioning of our used ConvNeXt accepts a colored segmentation map as input, and there is a color rgb(0,0,0) for a background/undefined category, which is not among the 150 ADE20K classes, so we find it suitable to use all-zero colormaps as the conditioning and regard the ControlNet as only receiving meaningful conditions from texts. While adapting, we again freeze the VAE encoders and text embedder, while training the denoising UNet, along with the ControlNet branch plugged in. For the task model SegFormer-B5, the input size is $512 \times 512$, so we resize the shorter side of input images to 512 while keeping the aspect ratio for all methods involved. As for our DUSA, the ControlNet requires an input size of $512 \times 512$. To get the most of semantic priors from diffusion models, we apply a sliding strategy to both ControlNet input (which is a non-square input with a 512 short side) and SegFormer **output**, which aligns the task model forward with compared methods, and thus the sliding on one image is finished in two steps. Note that SegFormer-B5 has a $4\times$ downsample while ControlNet has a $8\times$ downsample, therefore the logits of SegFormer-B5 can be larger than the latent space of ControlNet, so we further perform a $2\times$ downsample on the logits to prepare for the aggregation of noise predictions in Eq. (14).

As we resort to the conventional **text-to-image** formulation of diffusion models, a whole noise estimation map is predicted each time a condition is given. This is much different from classification as the segmentation results are at the pixel level, making an image-level candidate class selection non-trivial. Note that LogitNorm is still present before selection, which is applied to the logits in the channel dimension. We make a little modification to the selection strategy here, instead of attempting to change the structure of diffusion models as done in [32], to prove the versatility of DUSA. Specifically, we allow a budget of 20 classes for each input image, which is also split into a task model-based $top$-1 selection budget and a random budget, inheriting the spirits of DUSA for classification. We first gather the unique set of $top$-1 predicted classes from all pixels. If the number of gathered classes already exceeds the budget, we randomly suppress the redundant classes. Otherwise, if there is a surplus in the budget, we further perform a random selection from the remaining classes until the threshold is reached. The subsequent steps are the same as in classification.

### F.4 More Details on Compute Resources

We use Nvidia A6000 GPU with 48GB memory for all our experiments. For faster training, we use Automatic Mixed Precision with autocasts to fp16 for both classification and semantic segmentation. Additionally for semantic segmentation, gradient checkpointing is enabled for the task model. For test-time adaptation on classifiers with a batch size of 8, our DUSA takes around 43GB of memory and 1.5h training time for a single task (roughly 0.11s per image), while our DUSA-U takes around 28GB of memory and 1h training time for a single task (roughly 0.07s per image). For test-time semantic segmentation with a batch size of 1, we experiment on two Nvidia A6000 GPUs, with a total of 24GB+44GB=68GB and 2.5h training time for a single task (roughly 4.5s per image). As we perform three independent runs for each task, a total of 14 GPU days are required for results in

Table 1, 3 GPU days for Table 2, and 5 GPU days for Table 3, accumulating into around 22 A6000 GPU days for our main results. Preliminary experiments make the full research project require more compute than reported, but it's non-trivial for us to benchmark them all.

## G    Detailed Ablation Results on DUSA Components

We provide the detailed results in Table 4 for ResNet-50 and ConvNeXt-L on the three variants of corruptions in the Noise category in Table 5, namely Gaussian noise, Shot noise and Impulse noise, along with Pixelate corruption in the Digital category for ConvNeXt-L.

Table 5: Ablation on critical components in DUSA. Components in colored rows are not carried over to subsequent rows. Task-level results are provided for the Noise category.

| Variants | $k$ | $m$ | D.F. | D.B. | ResNet-50 (GN) | | | ConvNeXt-L (LN) | | | |
| --- | --- | --- | --- | --- | --- | --- | --- | --- | --- | --- | --- |
| | | | | | Gauss. | Shot | Impul. | Gauss. | Shot | Impul. | Pixel |
| Source-only | 0 | 0 | 0 | 0 | 22.1 | 23.0 | 22.0 | 56.7 | 56.2 | 58.3 | 42.3 |
| + score priors inspired loss (4) | 4 | 0 | 4 | 0 | 23.5 | 26.7 | 27.8 | 31.1 | 48.1 | 53.1 | 9.2 |
| + LogitNorm (4) | 4 | 0 | 4 | 0 | 42.5 | 45.3 | 44.2 | 59.3 | 61.7 | 61.7 | 9.6 |
| + adapt diffusion (4) | 4 | 0 | 4 | 4 | 40.5 | 42.4 | 40.8 | 59.0 | 59.1 | 55.4 | 49.3 |
| + LogitNorm (4) | 4 | 0 | 4 | 4 | 45.4 | 47.4 | 46.3 | 64.0 | 65.5 | 65.5 | 70.4 |
| + LogitNorm (6) | 6 | 0 | 6 | 6 | 45.4 | 47.6 | 46.5 | 64.0 | 65.6 | 65.6 | 70.7 |
| + uniform select (6) | 4 | 2 | 6 | 6 | 45.0 | 47.3 | 46.1 | 64.2 | 65.7 | 65.4 | 70.7 |
| + multinomial select (6) (DUSA) | 4 | 2 | 6 | 6 | 45.2 | 47.3 | 46.4 | 64.2 | 65.6 | 65.5 | 70.8 |
| + null conditioning (6) (DUSA-U) | 4 | 2 | 7 | 1 | 45.1 | 47.2 | 46.1 | 63.7 | 65.3 | 65.1 | 70.5 |

## H    Ensembling Timesteps in DUSA

In DUSA, we formulate our approach to extract knowledge from a single timestep, thereby enhancing the efficiency of adaptation. However, it is intriguing to investigate whether an ensemble of multiple timesteps would further improve performance. We experiment on ConvNeXt-L and present our findings in Table 6. The results indicate that, while ensembling timesteps does provide benefits, the performance gains may not be substantial enough to justify the increased computational overhead.

Table 6: Effects of ensembling timesteps in our DUSA. Experiments were conducted across four typical scenarios that fall into four main categories in ImageNet-C.

| Timestep(s) | Gauss. | Defoc. | Snow | Contr. |
| --- | --- | --- | --- | --- |
| {50} | 64.0 | 50.8 | 69.5 | 69.3 |
| {100} | 64.2 | 54.7 | 70.1 | 68.9 |
| {200} | 63.4 | 55.1 | 69.8 | 66.6 |
| {50,100} | **64.3** | 54.0 | **70.2** | 69.2 |
| {50,100,200} | **64.3** | **55.4** | **70.2** | 69.1 |

## I    Visualization of Test-time Semantic Segmentation Results

We visualize the test-time semantic segmentation results of our DUSA and compared methods in Fig. 5. A model checkpoint is saved after test-time adaptation over a whole corrupted ADE20K validation set. The checkpoint is then used to yield segmentation maps. We show results from four main categories of corruption, and segmentation maps are colorized with the ADE20K palette for better visual effects. Our DUSA, which exploits the structured semantic priors underneath the score-based diffusion model, shows superior capability in correcting erroneous predictions and providing fine-grained segmentation results.

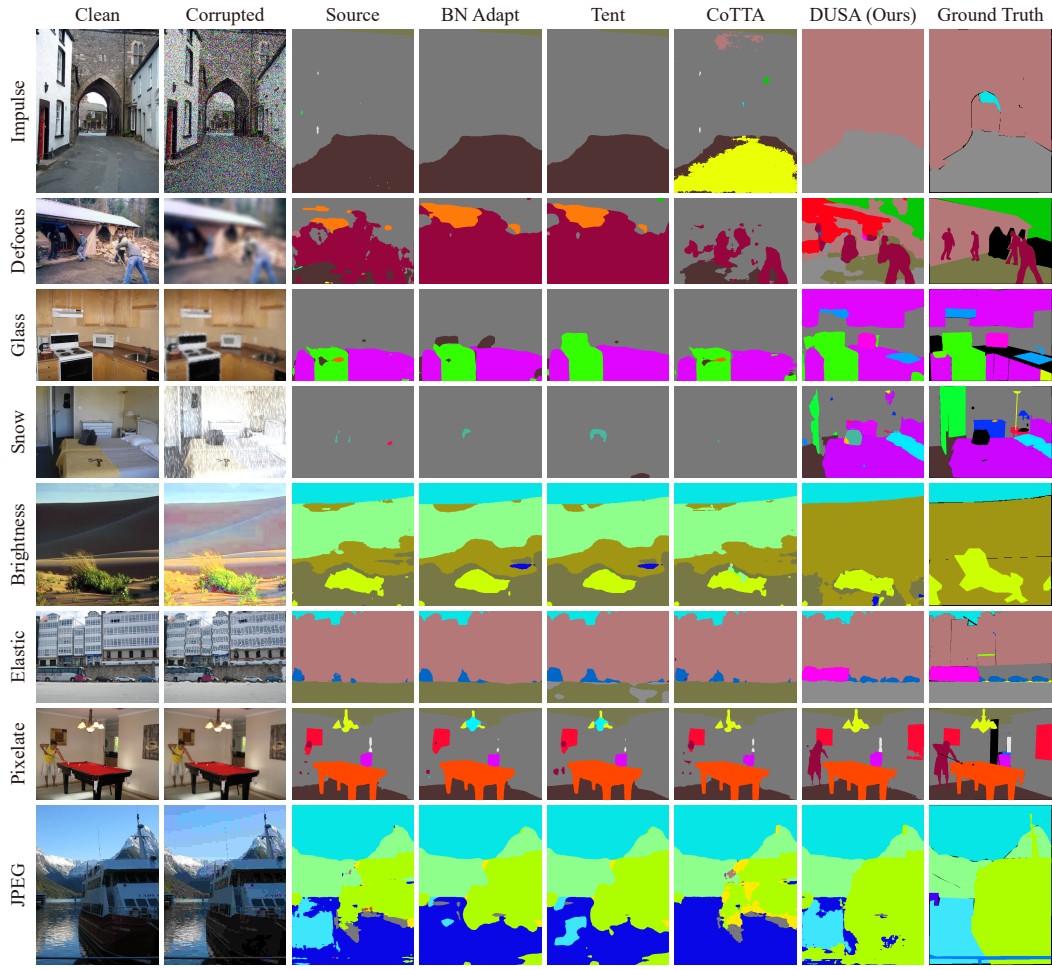

Figure 5: Visualization of test-time semantic segmentation results on ADE20K-C. From left to right: clean image from ADE20K, corrupted version of the image, results from source model, BN Adapt, Tent, CoTTA, our DUSA, and lastly the ground truth. DUSA results exhibit a favorable visual effect.

