# OpenReview forum: "Exploring Structured Semantic Priors Underlying Diffusion Score for Test-time Adaptation"
_NeurIPS.cc/2024/Conference — NeurIPS 2024 poster_

### Official Review · Reviewer_Your · 2024-07-10

**Soundness:** 3
**Presentation:** 3
**Contribution:** 3
**Rating:** 7
**Confidence:** 3

**Summary:**

The paper proposes DUSA, which uses diffusion models for test-time adaptation. Class codes are chosen for an image based on the results of a pretrained classifier and random sampling; these codes are used to generate noise candidates, which are combined based on the predicted probabilities for the associated classes. Error and weight updates are computed for both the classifier and the diffusion model based on the error between the aggregated and ground truth noises. This takes advantage of the semantics of the diffusion model to handle corrupted, challenging images not only for classification, but also for segmentation. The results include a variety of tasks and classifiers.

**Strengths:**

[S1] Figure 1 is an outstanding, very well-designed portrayal of the method.

[S2] The method is novel, with its use of single time steps and CSM.

[S3] The results are outstanding, with large improvements across multiple tasks, reproducible on multiple models and with hyperparameter settings that do not seem too brittle.

**Weaknesses:**

[W1] Existing art, e.g. [1], show there may be some benefit to using a few time steps. It also seems that in the image classification regime, the increase in computational complexity is unacceptably large. However, given existing work in the TTA area already uses many steps, it would have been nice to see some exploration of the impact of using more than 1 step, insofar as it would likely make the approximation in Equation 9 more accurate.

[W2] Considering the related work is positioned at the end, the introduction should do a little more to introduce and motivate the task. TTA is substantially more niche than other tasks, and the paper's narrative flow suffers from not making the motivation more explicit.





[1] Mukhopadhyay et al., Do text-free diffusion models learn discriminative visual representations?

**Questions:**

What is the latency of the method? Limitations mentions that it would not work for real time (e.g. autonomous driving) applications, but how far off is it?

Was any exploration of ensembling time steps attempted?

**Limitations:**

Yes.

---

> ### Author Rebuttal · Authors · 2024-08-07
>
> **Q1: Existing art, e.g. [1], show there may be some benefit to using a few time steps. It would have been nice to see some exploration of the impact of using more than 1 step, insofar as it would likely make the approximation in Equation 9 more accurate. Was any exploration of ensembling time steps attempted?**
>
> **A1:** Thanks for the valuable advice. The inspiring work [1] unveils the discriminativeness of U-Net features in diffusion models and achieves superior unsupervised unified representation learning performance across diverse tasks, where ensembling of timesteps is found beneficial. Our DUSA shares a similar spirit of exploring the discriminativeness in generative diffusion models, thus the ensembling of timesteps is definitely worth trying out.
>
> To explore the impact of ensembling timesteps as suggested, we experiment by performing DUSA on more than one timestep simultaneously and averaging their losses, and the results are as follows:
> |Timestep(s)|Gauss.|Defoc.|Snow|Contr.|
> |-|:-:|:-:|:-:|:-:|
> |{50}|64.0|50.8|69.5|69.3|
> |{100}|64.2|54.7|70.1|68.9|
> |{200}|63.4|55.1|69.8|66.6|
> |{50,100}|**64.3**|54.0|**70.2**|69.2|
> |{50,100,200}|**64.3**|**55.4**|**70.2**|69.1|
>
> It can be seen that, the ensembling of different timesteps **does bring a further improvement in performance**, thanks to the knowledge from multiple timesteps. Furthermore, our DUSA has its special advantage of draining knowledge from every single timestep, and can still get good results with reduced computational overhead. We appreciate the insightful advice and will add the discussions in the revision.
>
> [1] Mukhopadhyay et al., Do text-free diffusion models learn discriminative visual representations?
>
> **Q2: Considering the related work is positioned at the end, the introduction should do a little more to introduce and motivate the task.**
>
> **A2:** Thanks for the good suggestion. In test-time adaptation, a pre-trained model is updated on the fly to make accurate predictions on the incoming target samples without label access, which is challenging as the target data distribution might drift from that in pre-training. Being a competitive family of generative models, diffusion models exhibit great capability in modeling data distributions and even show discriminative potential in learned features [1,2,3,4], making them a strong candidate to provide guidance for discriminative tasks. In this work, we aim to extract discriminative priors from diffusion models to facilitate the challenging task of test-time adaptation.
>
> We thank the reviewer's efforts in improving the readability of our manuscript and will further polish up the expression and integrate the above discussions into the introduction in the revision.
>
> [1] Mukhopadhyay et al., Do text-free diffusion models learn discriminative visual representations?
>
> [2] Zhang et al., Diffusionengine: Diffusion model is scalable data engine for object detection.
>
> [3] Zhao et al., Unleashing text-to-image diffusion models for visual perception.
>
> [4] Xu et al., Open-vocabulary panoptic segmentation with text-to-image diffusion models.
>
> **Q3: What is the latency of the method? Limitations mentions that it would not work for real time (e.g. autonomous driving) applications, but how far off is it?**
>
> **A3:** Thanks for the question. As shown by the results in the table below, our DUSA predicts at a rate of 0.11s/image for classification with Diffusion Transformer (DiT), which is **-99.6% in time** compared to Diffusion-TTA. For segmentation, the time is 4.5s/image for segmentation with ControlNet.
> |ConvNeXt-L|Time/Image (A6000)|Gauss.|Defoc.|Snow|Contr.|
> |-|:-:|:-:|:-:|:-:|:-:|
> |Diffusion-TTA|27.71s|59.3|50.3|64.7|50.5|
> |DUSA|**0.11s (-99.6%)**|**64.2 (+4.9)**|**54.7 (+4.4)**|**70.1 (+5.4)**|**68.9 (+18.4)**|
>
> The latency is largely related to the selection of the diffusion model as well as the task complexity. With this said, we can freeze the diffusion model (DUSA-Frozen) and only update the task model in DUSA, which will still provide decent performance on a handful of tasks with **further reduced latency** and nearly real time, and the results are as follows:
> |ConvNeXt-L|Time/Image (A6000)|Gauss.|Defoc.|Snow|Contr.|
> |-|:-:|:-:|:-:|:-:|:-:|
> |DUSA|0.11s|64.2|54.7|70.1|68.9|
> |DUSA-Frozen|**0.05s**|58.4|33.7|64.2|62.0|
>
> We will continue researching how to reduce the latency for discriminative tasks further while utilizing the diffusion model's strong power of generative modeling.

---

> > ### Comment · Reviewer_Your · 2024-08-12
> > **Score unchanged**
> >
> > Thanks for the clarifications. I think they improve the paper, and I think they make my initial rating more accurate. The paper is indeed technically sound, with high impact on TTA.

---

> > > ### Author Response · Authors · 2024-08-12
> > > **Thank you for the suggestions and positive feedback**
> > >
> > > Dear Reviewer Your,
> > >
> > > Thank you for your thoughtful feedback. We greatly appreciate your acknowledgment and find your suggestions invaluable for enhancing our paper. We hope our work will shed light on the evolving field of test-time adaptation.
> > >
> > > Best regards,
> > >
> > > Submission335 Authors.

---

### Official Review · Reviewer_6Z3D · 2024-07-13

**Soundness:** 3
**Presentation:** 3
**Contribution:** 3
**Rating:** 6
**Confidence:** 4

**Summary:**

This work extends the Diffusion-test time adaption framework of [26] where the diffusion loss is averaged over timesteps to a simpler and theoretically justified framework, where a single timestep of the diffusion model can extract the semantic priors from the generative model. This reduces the instability of the Monte Carlo-based likelihood estimation over the timesteps. The proposed approach is applied to  ImageNet classification on ConvNext and segmentation.

**Strengths:**

1. The work is well-written and well-motivated with self-contained justification for using a single timestep for extracting the semantic prior from the diffusion model.
2.  Extensive experiments on the full test-time adaptation and continual test-time adaptation of ImageNet show that the approach outperforms prior work by a large margin.
3. Adequate ablations are performed to show the effect of  selection of timestep and different components of the proposed framework.

**Weaknesses:**

1. During optimization updates are performed on the diffusion weights as well. How the joint update for the two networks is performed is unclear. The timestep selection should impact the diffusion model's update schedule and the output logits.
2. The computational overhead needed for the approach and the gain over the prior work are not discussed.
3.  The work claims that the approach can be applied to any discriminative task. Class embedding as conditional is generally required for the diffusion model. This should limit the discriminative settings that can be considered for TTA with diffusion models.
4. The paper does not discuss how are the conditionals computed for the segmentation task. The increase in computational overhead should also be discussed with the increased complexity of the task.
5. Single timestep for adapting diffusion models on vision tasks has been discussed in [a] and [b]. These works can be discussed in the related work.

[a] Unsupervised semantic correspondence using stable diffusion. NeurIPS 2023.

[b] Slime: Segment like me. ICLR 2024.

**Questions:**

1. How is the update performed for the diffusion model as in [26] updating the diffusion model is optional. Does it incur any additional overhead?
2. Do different tasks, for example, segmentation and classification, have a single timestep where they perform well, or depending on the granularity of the task the choice of the timestep is different?

**Limitations:**

The limitations of the work are discussed.

---

> ### Author Rebuttal · Authors · 2024-08-07
>
> **Q1: How the joint update for the two networks is performed is unclear.**
>
> **A1:** Thanks. Before adaptation, **a single fixed timestep is selected** and other timesteps are dismissed. Hereafter, the diffusion model is **only updated on this single timestep**, i.e., we added a **same** level of noise to all images and denoise them at a **same** timestep. Note that the task model predicts on clean images, thus **not affected by timestep**. The diffusion noise estimations and the task model predictions are then integrated into Eq. 10 for a joint update.
>
> **Q2: The timestep selection should impact the diffusion model's update schedule and the output logits.**
>
> **A2:** Thanks. In our DUSA, the diffusion model is only updated on a **single fixed timestep**. We agree that timestep selection should impact the amount of noise in $x_t$ and the timestep condition $t$ for noise estimation, but we argue that the timestep in DUSA is **fixed throughout adaptation** and how we select it makes no difference to DUSA's pipeline. Besides, the task model predicts logits from clean images and is **not affected by timestep selection**.
>
> **Q3: The computational overhead needed for the approach and the gain over the prior work are not discussed.**
>
> **A3:** Thanks. Firstly, with $K$ classes and $T$ diffusion timesteps, the prior work Diffusion-TTA has a computational complexity of $\mathcal{O}(T)$, indeed 180 timesteps for each image. By contrast, our DUSA has an $\mathcal{O}(K)$, further reduced to $\mathcal{O}(b)(b\ll K)$ by practical designs (lines 136-139, 193). We list the computation time with gain as follows:
> |ConvNeXt-L|Time/Image (A6000)|Gauss.|Defoc.|Snow|Contr.|
> |-|:-:|:-:|:-:|:-:|:-:|
> |Diffusion-TTA|27.71s|59.3|50.3|64.7|50.5|
> |DUSA|**0.11s (-99.6%)**|**64.2 (+4.9)**|**54.7 (+4.4)**|**70.1 (+5.4)**|**68.9 (+18.4)**|
>
> It can be seen that our DUSA shows significant gain over Diffusion-TTA with a tremendous reduction (-99.6%) in computational overhead.
>
> **Q4: Class embedding as conditional is generally required for the diffusion model. This should limit the discriminative settings that can be considered for TTA with diffusion models.**
>
> **A4:** Thanks. We agree that diffusion models generally require class embeddings. However, the specific form of class embeddings can vary. For classes with semantic meanings, we can construct a prompt containing class names and use a text model as embedder. For classes without explicit meanings, we can discretize them to IDs and map them to embedding vectors. In this way, we believe any discriminative task can be taken into consideration.
>
> **Q5: How are the conditionals computed for the segmentation task? The increase in computational overhead should also be discussed with the increased complexity of the task.**
>
> **A5:** Thanks. As we stick to text-to-image diffusion models, the conditionals for segmentation are computed **at class level**. Specifically, we construct a prompt "a photo of a {class}" and use the diffusion model's text encoder to obtain class embedding. Please note that **a single conditional noise estimation** in diffusion could also be viewed as **a noise estimation for every single pixel**. Therefore, the computation of conditionals for the segmentation task is **as simple as** in classification: given a class, predict the noise. The only difference is the way of ensembling noise estimations with predictions as in Eq. 12, which introduces little overhead.
>
> The increase in computational overhead is first raised by a large network to adapt in segmentation. Besides, the overhead is partially due to using a larger diffusion model (ControlNet, 1.7B params) than that in classification (DiT, 675M params) and the sliding-window strategy for non-square images. Lastly, the overhead also comes from the unpolished use of our CSM module for this task. We find that our method is versatile and competitive across tasks, and it's majorly the task complexity that adds to overhead. We will add these discussions in the revision.
>
> **Q6: Single timestep for adapting diffusion models on vision tasks has been discussed in [a] and [b]. These works can be discussed in the related work.**
>
> **A6:** Good suggestion. [a] gets strong results in finding semantic correspondences by obtaining transferable prompt embeddings at a single timestep. [b] achieves significant gains in the one annotation-based image segmentation task at any granularity by learning reusable text embeddings at a single timestep. While also using a single timestep, our DUSA focuses on utilizing the diffusion model to adapt a discriminative model. We find the works relevant and will discuss them in the revision.
>
> [a] Unsupervised semantic correspondence using stable diffusion. NeurIPS 2023.
>
> [b] Slime: Segment like me. ICLR 2024.
>
> **Q7: How is the update performed for the diffusion model as in [26] updating the diffusion model is optional. Does it incur any additional overhead?**
>
> **A7:** Thanks. As described in A1, we update the diffusion model on a single fixed timestep. Like [26], updating the diffusion model is optional in DUSA, and we show the performance of freezing the diffusion model in **Table R6** in uploaded PDF. It can be seen that when not updating diffusion model, our method still outperforms [26]. Besides, with updating, our DUSA takes the lead by a large margin, with a vastly reduced overhead of **-99.6%** adaptation time compared to [26], as shown in A3.
>
> **Q8: Do different tasks, for example, segmentation and classification, have a single timestep where they perform well, or depending on the granularity of the task the choice of the timestep is different?**
>
> **A8:** Thanks. We vary the selected timestep for segmentation and report the results in **Figure R2** in uploaded PDF. As the timestep being too large or too small is not preferred (lines 169-172), we focus on a narrower range here. It can be seen that there is no global best for all scenarios, but our $t=100$ proves effective across tasks.

---

> > ### Comment · Reviewer_6Z3D · 2024-08-12
> >
> > Thank you for the clarifications. The rebuttal discusses the listed weaknesses and there are no further questions.

---

> > > ### Author Response · Authors · 2024-08-12
> > > **Thank you for your positive feedback**
> > >
> > > Dear Reviewer 6Z3D,
> > >
> > > It's good to know that we have addressed your concerns. We really appreciate your thoughtful comments and response, and are sincerely grateful for your efforts in improving the quality of our work.
> > >
> > > Best regards,
> > >
> > > Submission335 Authors.

---

### Official Review · Reviewer_oxUu · 2024-07-16

**Soundness:** 3
**Presentation:** 3
**Contribution:** 3
**Rating:** 6
**Confidence:** 4

**Summary:**

This paper proposes to perform Test-Time Adaptation (TTA) with the help of diffusion models, based on the theoretical observations that effective discriminative priors are hidden within conditional diffusion losses.

The method involves a joint adaption of the task discriminative model and the generative diffusion model, with its optimization objective given from a theoretical perspective. To make it more efficient, the paper shows that the objective can be decoupled to fit modern classifier-free guidance (CFG) based models, and can still work well when only using one single timestep and a few selected class candidates. Moreover, the method is shown to naturally fit dense prediction tasks.

Experimental results on fully TTA and continual TTA tasks show the efficacy on both image classification and semantic segmentation, surpassing previous diffusion-based TTA methods.

**Strengths:**

**Novelty in research problem and method.** Utilizing the discriminative natures within generative diffusion models to enhance discriminative  task models is an interesting and underexplored topic, and could be valuable for the research community. Extracting the underlying semantic  knowledge from diffusion losses from a theoretical perspective is novel (Eq. 10). Empirical contributions to improve the efficiency and versatility are also valuable.

**Significant results.** The evaluations on two discriminative tasks under two different TTA settings show the effectiveness of the proposed method, which outperforms previous baselines by large margins, including diffusion-based ones. These empirical results are significant.

**Weaknesses:**

**Clarity on the training objective in Eq. 11.** Though the objective proposed in Eq. 10 is well-supported from the theoretical perspective, the alternative and more efficient one in Eq. 11 is somewhat counter-intuitive and needs further elucidation. Please refer to the "Questions" section.

**The role of diffusion models.** The proposed method (especially for image classification) relies on a conditional diffusion model that is pre-trained on the same image dataset and with the same set of class labels. Though the experiments on semantic segmentation are done with a large text-to-image model pre-trained on web images, the paper does not show whether it is better than a model pre-trained on the same distribution. Besides, how does the quality (e.g., sampling FID, model capacity, training duration, training dataset diversity) of diffusion models affect TTA performance?

The method also relies on the fine-tuning of diffusion models, rather than only evaluating off-the-shelf models. This makes the TTA process more costly, since diffusion models are usually much larger than discriminative task models.

**Questions:**

The reviewer has some questions about the proposed objective.
- Can you give some insights or intuitions on why the modified objective (Eq. 11) works as well as the original objective (Eq. 10)?
- Related to the above, why the unconditional part in Eq. 11 is still needed?
  - Firstly, the unconditional version of CFG-based models is much weaker than the conditional version, and they usually do not agree well with each other. Do the unconditional training really help the conditional noise estimation  $\epsilon_\phi(x_t,t,c_y)$ ?
  - Moreover, the conditional part in Eq. 11 does not involve tuning the diffusion weights $\phi$, then why does the diffusion model (unconditional version) still need to be constrained by a regular CFG objective, given that it has already been well pre-trained?

**Limitations:**

The authors have adequately discussed most of the limitations, and there have no concerns about negative societal impact.

---

> ### Author Rebuttal · Authors · 2024-08-07
>
> **Q1: Though Eq. 10 is well-supported from theoretical perspective, the alternative Eq. 11 needs further elucidation. Insights or intuitions on why the modified objective (Eq. 11) works as well as the original objective (Eq. 10)?**
>
> **A1:** Thanks for the valuable question. Please note that we can view Eq. 10 from two perspectives: (a) the task model learns from the diffusion model, (b) weighted optimization is performed so that $\epsilon_\phi(x_t,t,c_y)$ with a larger weight $p_\theta(y|x_0)$ is more responsible for estimating true noise. Therefore, Eq. 10 **encourages the conditional diffusion model to adapt to test-time data**, under **instruction of task model predictions**.
>
> Similarly, two perspectives of Eq. 11 can be shown. The conditional part adapts the task model as in Eq. 10. For the unconditional part, we start from Eq. 9:
> $$\epsilon\approx\sum_yp(y|x_t)\epsilon_\phi(x_t,t,c_y).\quad(\text{Eq. 9})$$
> Without loss of generality, we ignore the estimation error below. In a CFG-based diffusion model, the true noise in Eq. 9 can be unconditionally estimated:
> $$\epsilon_\phi(x_t,t,\varnothing)=\sum_yp(y|x_t)\epsilon_\phi(x_t,t,c_y).\quad(1)$$
> Please note that all noise estimations are from **a same diffusion model**, and the weights $p(y|x_t)$ are the **implicit priors in the diffusion model**. Therefore, we can view (1) as an **internal constraint** of the diffusion model. Intuitively, the unconditional part **implicitly enforces conditional adaptation of the diffusion model to test-time data**, now under **instruction of the implicit priors from the diffusion model itself**, explaining why Eq. 11 works as well as Eq. 10. We will add the discussions in the revision.
>
> **Q2: The unconditional version of CFG-based models is much weaker than the conditional version, and they usually do not agree well with each other. Do the unconditional training really help the conditional noise estimation?**
>
> **A2:** Thanks. We agree that the unconditional version is weaker and there can be disagreement. However, this version is still decent for noise estimation, as it also undergoes extensive diffusion training. Besides, in A1 we show that **the unconditional training helps adapt conditional noise estimations to test-time data** in CFG-based models. Therefore, we prefer to view the two versions as **cooperative**, and their agreement is not forced.
>
> We compare unconditional training with freezing the diffusion model in **Table R1** in uploaded PDF. We can see that the unconditional results are comparable to conditional training. This strengthens our belief that **the unconditional training does help the conditional noise estimations**, with reduced overhead.
>
> **Q3: The conditional part in Eq. 11 does not involve tuning the diffusion weights, why does the diffusion model (unconditional version) still need to be constrained by a regular CFG objective?**
>
> **A3:** Thanks. The regular CFG objective **explicitly enhances both conditional and unconditional noise estimations** for better generation. In contrast, Eq. 11 focuses on utilizing the diffusion model to **facilitate task model adaptation** (conditional part) and leveraging the implicit priors to **adapt the conditional noise estimations** (unconditional part), as shown in A1.
>
> We constrain the diffusion model with unconditional training to handle a **potential distribution shift** between diffusion model's training set and test-time data. Results are shown in **Table R1** in uploaded PDF. If the diffusion model is generalizable to unseen data, unconditional training can be removed with limited sacrifice in performance (DUSA-Frozen for Gaussian). Otherwise, should the test-time data be out-of-distribution for the diffusion model, the absence of unconditional training might cause degradation (DUSA-Frozen for Defocus). Therefore, the constraints are preferred for a more consistent performance gain.
>
> **Q4: Whether a large text-to-image model pre-trained on web images is better than a model pre-trained on the same distribution for segmentation.**
>
> **A4:** Thanks. Actually, the diffusion model we adopt for semantic segmentation is a ControlNet (based on Stable Diffusion v1.5) fine-tuned on ADE20K, same distribution as the task model. Although not equal to training from scratch, we assume the fine-tuning makes the diffusion model aware of the task model's training data distribution.
>
> We compare the results of adapting with ControlNet against Stable Diffusion models in **Table R3** in uploaded PDF. It can be seen that the diffusion model's training on the same distribution as the task model **generally merits more performance gain**, as it narrows the gap between the task model and the diffusion model for better cooperation.
>
> **Q5: How does the quality of diffusion models affect TTA performance?**
>
> **A5:** Thanks. We compare the quality of popular diffusion models in **Table R4** in uploaded PDF. We apply our DUSA to them and show classification in **Table R2** and segmentation in **Table R3** in uploaded PDF. It can be seen that while a diffusion model of better quality fosters TTA performance (SD series), one training on the same distribution as the task model is generally more favorable (blue lines).
>
> **Q6: The method also relies on the fine-tuning of diffusion models, rather than only evaluating off-the-shelf models. This makes the TTA process more costly, since diffusion models are usually much larger than discriminative task models.**
>
> **A6:** Thanks. We agree that the fine-tuning of diffusion models makes TTA more costly, but we highlight the substantial performance gain obtained by updating the diffusion model (DUSA) against freezing it (DUSA-Frozen), as shown in **Table R1** in uploaded PDF.
>
> Besides, our DUSA is **more than 100x faster** in comparison with Diffusion-TTA, and approaches the strong traditional TTA method EATA in speed. The results are in **Table R5** in uploaded PDF, where our DUSA shows leading performance with little extra cost.

---

> > ### Comment · Reviewer_oxUu · 2024-08-13
> >
> > Thank you for the detailed explanations, and these have addressed most of my concerns. Therefore, I lean towards acceptance, and I encourage the authors to add the discussions about the unconditional / conditional parts in Eq. 11 to the main paper.
> >
> > The points that "unconditional part to handle a potential distribution shift between training and test-time data" and "conditional part to guide the task model to learn from the diffusion model" are interesting.

---

> > > ### Author Response · Authors · 2024-08-13
> > > **Thanks for the response**
> > >
> > > Dear Reviewer oxUu,
> > >
> > > Thank you for the positive feedback. We find your suggestions and questions really helpful in improving our work. We will follow your advice and add the interesting discussion on unconditional / conditional parts in Eq. 11 to the main paper in the revision.
> > >
> > > Best regards,
> > >
> > > Submission335 Authors.

---

### Official Review · Reviewer_V2Wk · 2024-07-17

**Soundness:** 3
**Presentation:** 2
**Contribution:** 2
**Rating:** 6
**Confidence:** 5

**Summary:**

Tackling the limitations of prior research on test-time adaptation using pre-trained diffusion models, this study expands the diffusion prior for more practical scenarios. The authors extend the use of the diffusion prior to dense prediction tasks, enhancing inference speed with a refined diffusion loss function that eliminates the need for time averaging. This time-agnostic characteristic of the proposed diffusion loss undergoes rigorous validation through mathematical derivations and straightforward experimental validations.

While the pre-trained diffusion prior proves effective in various tasks, this study also demonstrates its utility in test-time adaptation for both classification and semantic segmentation with remarkable performance gain. These advancements are particularly crucial for practitioners in the field, offering more efficient and robust tools for real-time applications.

**Strengths:**

## Test-Time Adaptation for Dense Prediction Tasks

Recent advancements have seen the extensive use of large-scale pre-trained diffusion models as priors for various tasks. The application of diffusion priors in test-time adaptation for dense prediction tasks is a particularly intriguing discovery and is deemed to be a valuable approach moving forward.

## Time-Step Efficiency of the Proposed Method

Traditional diffusion models often require generating multiple samples due to the expectation across temporal steps in their loss functions. This research successfully removes the expectation over temporal steps, thereby enhancing efficiency. However, the formulation of the proposed method isn't fully understood, prompting additional questions.

**Weaknesses:**

## Misleading Title of the Paper

The algorithm described in the Appendix should ideally be integrated into the main body of the paper.
Ambiguity in the Use of the Adjective 'Fresh'

## the term "fresh" is used ambiguously in lines 66-68:
> A fresh proposition is provided from a theoretical perspective to extract discriminative priors from score-based diffusion models, which are capable of handling both classification and dense prediction tasks at test time in a single timestep.

**Questions:**

## Accuracy of Diffusion-TTA with a Single Time Step as in Figure 3

The reduction of necessary time steps from an average of 180 to a dramatically lower number is a significant advantage. However, the verification of this reduction and a clear comparison with standard diffusion-TTA remain unclear. Specifically, it is questioned whether averaging out the loss function $\mathcal{L}_{DUST}(\theta, phi; t)$ over T=180 under the same conditions as TTA-Diffusion would affect performance. Conversely, the performance of TTA-Diffusion with only a single timestep, as shown in Figure 3, is also in question.

## Formal Proof of Biased Approximation

Diffusion-TTA relies heavily on the Monte Carlo method across up to 180 timesteps, resulting in a biased approximation of likelihood and high computational complexity. This methodological reliance is confusing since Monte Carlo estimations generally converge towards more accurate values with more samples. The paper mentions a biased approximation but lacks concrete proof. It's questioned whether the proposed DUST method can demonstrate that it does not rely on biased approximations; if not, such claims might be considered exaggerated.

## Advantages Over Diffusion-TTA in Dense Prediction Tasks

There are doubts about whether Diffusion-TTA is also applicable and effective for dense prediction tasks, which is missing in the manuscript.

## Effect of Training Set on Diffusion Prior

It would be interesting to see the effects of using a diffusion model trained on ImageNet instead of one trained on a large dataset like Stable Diffusion for an apple-to-apple comparison. Given that Stable Diffusion benefits from a larger dataset, not comparing it to other models might be seen as a limitation. Despite the inevitable nature of research progression, the lack of analysis on this aspect in both the current paper and TTA-Diffusion is questioned. The potential for performance improvement remains, considering the model's training on noise through diffusion.

**Limitations:**

The limitations of the paper have been highlighted within the weaknesses section, with detailed questions addressed above.

---

> ### Author Rebuttal · Authors · 2024-08-07
>
> **Q1: The algorithm in Appendix should be integrated into the main paper.**
>
> **A1:** Thanks. Following your advice, we will integrate the algorithm into the main body of the paper in the revision.
>
> **Q2: The term "fresh" is used ambiguously in lines 66-68.**
>
> **A2:** Good suggestion. We agree that "a fresh proposition" might be ambiguous, and we clarify by restating it as "a novel proposition" and will update it in the revision.
>
> **Q3: The verification of this reduction and a clear comparison with standard Diffusion-TTA remain unclear. Whether averaging out the loss function $\mathcal{L}_{DUSA}(\theta, \phi; t)$ over T=180 under the same conditions as Diffusion-TTA would affect performance.**
>
> **A3:** Thanks. Diffusion-TTA is built on the Monte Carlo estimation of likelihood, and demands **randomly sampling up to 180 timesteps** to boost performance. By contrast, our DUSA utilizes semantic priors from a single timestep, and needs only a **single fixed timestep** to achieve superior performance.
>
> For a clear comparison with the standard Diffusion-TTA, **we put our DUSA under the same conditions of Diffusion-TTA**: (a) we change the timestep selection **from fixed to random sampling**, (b) we increase the timesteps number **from 1 to 180**. The comparison results are as follows:
> |ConvNeXt-L|Gauss.|Defoc.|Snow|Contr.|
> |-|:-:|:-:|:-:|:-:|
> |Diffusion-TTA|59.3|50.3|64.7|50.5|
> |DUSA|**64.0 (+4.7)**|**54.7 (+4.4)**|**69.9 (+5.2)**|**68.0 (+17.5)**|
>
> It can be seen that our DUSA significantly outperforms Diffusion-TTA even under its same conditions.
>
> **Q4: Comparision of Diffusion-TTA and DUSA with only a single timestep, as shown in Figure 3.**
>
> **A4:** Thanks. To compare under only a **single fixed** timestep, **modifications are made to Diffusion-TTA**: (a) we reduce the utilized number of timesteps **from 180 to 1**, (b) we change the timestep selection **from random sampling to a fixed timestep**. The comparison results for $t=100$ are as follows:
> |ConvNeXt-L|Gauss.|Defoc.|Snow|Contr.|
> |-|:-:|:-:|:-:|:-:|
> |Diffusion-TTA|59.8|48.1|64.3|61.5|
> |DUSA|**64.2 (+4.4)**|**54.7 (+6.6)**|**70.1 (+5.8)**|**68.9 (+7.4)**|
>
> Our DUSA surpasses Diffusion-TTA by a large margin, justifying the superiority of DUSA on only a single timestep. More results on varied timesteps can be found in **Figure R1** in uploaded PDF.
>
> **Q5: The paper mentions a biased approximation but lacks concrete proof. Whether the proposed DUSA method can demonstrate that it does not rely on biased approximations.**
>
> **A5:** Thanks. Indeed, our original claim "... for a biased approximation of likelihood" may cause confusion, and a better expression is "... to estimate a biased approximation of likelihood". Actually, the bias **comes from Diffusion-TTA's theoretical approximation of the likelihood** instead of the Monte Carlo method, and we provide a proof below.
>
> Diffusion-TTA is built on the evidence lower bound (ELBO) of log-likelihood in diffusion:
> $$\log p_\phi(x_0|c)\geq \mathbb{E}\_q\Big[\log\frac{p\_\phi(x\_{0:T}|c)}{q(x\_{1:T}|x_0)}\Big],\quad(1)$$
> where $q$ is the forward process and $p_\phi$ is the backward process. We can further derive the ELBO in (1) with denoising network $\epsilon_\phi$:
> $$\mathbb{E}_q\Big[\log\frac{p\_\phi(x\_{0:T}|c)}{q(x\_{1:T}|x_0)}\Big]=-\mathbb{E}\_\epsilon\Big[\sum\_{t=2}^Tw_t\\|\epsilon-\epsilon\_\phi(x_t,t,c)\\|_2^2-\log p\_\phi(x_0|x_1,c)\Big]+C\approx-T\mathbb{E}\_{\epsilon,t}[\\|\epsilon-\epsilon\_\phi(x_t,t,c)\\|_2^2]+C,\quad(2)$$
> where the **theoretical approximation** is made by simplifying weights $w_t$ to 1 and ignoring the $\log p\_\phi(x_0|x_1,c)$ term. Note that Diffusion-TTA actually estimates the simplified objective in (2), thus the result is **theoretically biased** from real likelihood because the approximation in (2) just prevents the equality sign in (1) from holding.
>
> In contrast, our DUSA objective in Eq. 10 is built on the theory in Eq. 4 with **no theoretical approximation**. The task model thus directly utilizes semantic priors from the diffusion model. We will add the discussions in the revision.
>
> **Q6: Advantages over Diffusion-TTA in dense prediction tasks. Whether Diffusion-TTA is also applicable and effective for dense prediction tasks.**
>
> **A6:** Thanks. Diffusion-TTA is applicable to dense prediction tasks, as depicted in [26]. However, a pixel-level conditioning capability is assumed on the diffusion model for Diffusion-TTA to be effective, which requires re-training of diffusion models and reduces versatility. We re-train the diffusion model on ADE20K according to [26] to fit segmentation task, and comparison results are as follows:
> |SegFormer-B5|Gauss.|Defoc.|Snow|Contr.|
> |-|:-:|:-:|:-:|:-:|
> |Diffusion-TTA|15.3|23.5|23.7|22.9|
> |DUSA|**23.6 (+8.3)**|**24.7 (+1.2)**|**27.3 (+3.6)**|**27.1 (+4.2)**|
>
> It can be seen that our DUSA consistently outperforms Diffusion-TTA on the dense prediction task.
>
> **Q7: The effects of using a diffusion model trained on ImageNet instead of one trained on a large dataset like Stable Diffusion. Given that Stable Diffusion benefits from a larger dataset, not comparing it to other models might be seen as a limitation.**
>
> **A7:** We appreciate the reviewer's insights in exploring our DUSA on diffusion models varying in training set. We would clarify that, for image classification, an ImageNet pre-trained Diffusion Transformer (DiT) is already used throughout our paper. We only use Stable Diffusion-based ControlNet for the segmentation task. As suggested, we further compare to other diffusion models and provide results for classification in **Table R2** in uploaded PDF and segmentation in **Table R3** in uploaded PDF.
>
> We can find that a diffusion model trained on the same distribution as the task model is generally more beneficial for adaptation. Intuitively, the diffusion model's being aware of the pre-training data of the task model eliminates the disagreement between them, fostering their cooperation under a narrowed distribution gap.

---

> > ### Comment · Reviewer_V2Wk · 2024-08-10
> >
> > **Regarding all questions except Question 5**
> >
> > Thank you for addressing my concerns related to the experiments. Your comprehensive work, especially given the short timeframe, has completely resolved my questions.
> >
> > **Regarding Question 5**
> >
> > I am still unclear on a particular point. While I understand that Diffusion-TTA relies on a biased approximation due to the optimization of the ELBO, I’m confused about the biased approximation mentioned in Eq (7), even though the current formulation begins from Eq (4). It seems to me that this could be considered a form of circular reasoning since the denoising estimator in Eq (7) is trained using the ELBO. Is my interpretation correct?

---

> > > ### Author Response · Authors · 2024-08-11
> > > **Further response to concerns on approximation bias**
> > >
> > > Dear Reviewer V2Wk,
> > >
> > > We are glad to hear that your concerns related to the experiments have been completely addressed. We would like to make a further demonstration of the **unbiased estimation** in our DUSA. As a short answer, the denoising estimator in Eq (7) is actually **unbiased**, there is **no theoretical approximation** for Eq (7), and we argue that **no circular reasoning** exists in our DUSA. Please find the details below.
> > >
> > > In Eq (7), we show that the true noise $\epsilon$ equals the weighted ensemble of conditional score functions $\nabla_{x_t}\log p(x_t|y)$ **with no approximation**. The reviewer's confusion might come from our **Eq (8)**, where the conditional score functions $\nabla_{x_t}\log p(x_t|y)$ are further **estimated** by conditional noise estimations $\epsilon_\phi(x_t,t,c_y)$ from the diffusion model.
> > >
> > > We would like to clarify that, the approximation symbol $\approx$ in Eq (8) **does not indicate a bias in estimation**, but rather indicates that the conditional noise estimations **might not be totally accurate** as they are predictions from the denoising network in diffusion. Indeed, **the denoising estimators are unbiased**, as they are trained by the simplified objective in Eq (3). To train a diffusion model with Eq (3), we first randomly sample noise $\epsilon\sim\mathcal{N}(0,I)$ and uniformly sample timestep $t$, and obtain noised sample $x_t$ following Eq (2). Then, the training objective in Eq (3) is optimized by minimizing:
> > > $$\mathcal{L}(\phi)=\mathbb{E}\_{x_t|\epsilon,t}[\\|\epsilon-\epsilon_\phi(x_t,t,c_y)\\|_2^2].$$
> > >
> > > At the optimal point of the objective, we have:
> > > $$\frac{\partial\mathcal{L}(\phi)}{\partial\epsilon_\phi}=\mathbb{E}\_{x_t|\epsilon,t}[2(\epsilon-\epsilon_\phi(x_t,t,c_y))]=0,$$
> > > which implies:
> > > $$\mathbb{E}\_{x_t|\epsilon,t}[\epsilon_\phi(x_t,t,c_y)]=\mathbb{E}\_{x_t|\epsilon,t}[\epsilon]=\epsilon,\quad\text{Eq}~(*)$$
> > > meaning that the conditional noise estimations $\epsilon_\phi(x_t,t,c_y)$ are **unbiased** with regard to the true noise $\epsilon$.
> > >
> > > For a more comprehensive understanding of the **unbiased estimation** in our DUSA, we provide a step-by-step explanation below, justifying that **there is no circular reasoning**.
> > >
> > > 1. We start from the **equation** Eq (4), where the unconditional score function $\nabla_x\log p(x)$ can be expressed by an ensemble of conditional score functions $\nabla_x\log p(x|y)$.
> > > 2. With Eq (6) in Corollary 1, we show that there is a **equation** between the score function and the true noise $\epsilon$ on every single timestep.
> > > 3. We now replace the unconditional score function in Eq (4) with Eq (6) and get the **equation** in Eq (7).
> > >
> > > We remind that **no approximation is made** during all steps above. Also, Corollary 1 indicates that:
> > > $$\nabla_{x_t}\log p(x_t|y)=-\frac{\epsilon}{\sqrt{1-\bar{\alpha}_t}},\quad\text{Eq}~(**)$$
> > > as the true noise $\epsilon$ is sampled in the forward process and thus not relevant to the existence of conditioning. The proof for conditional score functions resembles that for Corollary 1 in Appendix C.
> > >
> > > 4. We then use conditional noise estimations $\epsilon_\phi(x_t,t,c_y)$ to **estimate** the true noise $\epsilon$ in Eq $(**)$, where the estimations are all **unbiased**, as proved in Eq $(*)$. This step results in Eq (8).
> > > 5. Combining Eq (7) and Eq (8), we instantly get Eq (9) and thus our DUSA objective in Eq (10). This step involves **no** approximation or estimation.
> > >
> > > Through this **non-circular** chain of reasoning, we can find that in DUSA there is **no theoretical approximation** and **the estimations are all unbiased**.
> > >
> > > We hope the proofs and chain of reasoning above have addressed the theoretical concerns. Please let us know if you have further questions or suggestions, and we are more than happy to continue the discussion.
> > >
> > > Best regards,
> > >
> > > Submission335 Authors.

---

> > > > ### Comment · Reviewer_V2Wk · 2024-08-13
> > > >
> > > > Thank you for the detailed explanations, and I apologize for the confusion. I was actually referring to the approximation in Eq (8), not Eq (7). You have made that point clear.
> > > >
> > > > The point of confusion lies in the fact that there is no guarantee that the conditional noise estimator is unbiased under my current understanding. More clearly, Eq (*) in the comment shows that the expectation over the noise estimator leads to $\epsilon$, not $0$. It seems that an additional expectation over the noise is required to claim the unbiased estimator. Am I wrong?
> > > >
> > > > Thank you for your effort and dedication for thoughtful discussions.

---

> > > > > ### Author Response · Authors · 2024-08-13
> > > > > **Further response to concerns on unbiased conditional noise estimator**
> > > > >
> > > > > Dear Reviewer V2Wk,
> > > > >
> > > > > Thank you for the further discussion. We would like to remind that, according to the definition, the bias of an estimator is calculated as **the difference between the expected value of the estimator and the real value to estimate**. In our DUSA, the true noise $\epsilon$ is estimated by $\epsilon_\phi(x_t,t,c_y)$, therefore with the proof in Eq $(*)$ the bias should be calculated by:
> > > > >
> > > > > $$\text{Bias}(\epsilon_\phi(x_t,t,c_y),\epsilon)=\mathbb{E}\_{x_t|\epsilon,t}[\epsilon_\phi(x_t,t,c_y)]-\epsilon=0,$$
> > > > >
> > > > > which means the conditional noise estimator $\epsilon_\phi(x_t,t,c_y)$ is indeed an **unbiased estimator** of the true noise $\epsilon$.
> > > > >
> > > > > We hope the explanations have resolved the concerns. We are more than happy to discuss if you have further questions.
> > > > >
> > > > > Best regards,
> > > > >
> > > > > Submission335 Authors.

---

> > > > > > ### Comment · Reviewer_V2Wk · 2024-08-13
> > > > > >
> > > > > > Thank you for the clarification. I now have a better understanding of the derivation in the main paper, particularly regarding the temporal and noise terms, which had previously caused some confusion. I have updated my rating from 5 to 6.
> > > > > >
> > > > > > Thank you for your effort and dedication to rebuttal and discussion.

---

> > > > > > > ### Author Response · Authors · 2024-08-13
> > > > > > > **Thank you for your efforts and positive feedback**
> > > > > > >
> > > > > > > Dear Reviewer V2Wk,
> > > > > > >
> > > > > > > Thank you very much for your efforts in helping us enhance the rigor of our work. We truly appreciate the discussion, and we find your suggestions and in-depth questions extremely helpful in strengthening our paper.
> > > > > > >
> > > > > > > Best regards,
> > > > > > >
> > > > > > > Submission335 Authors.

---

### Author Rebuttal · Authors · 2024-08-07

We sincerely thank all reviewers for their efforts in reviewing our paper and the constructive feedbacks that are quite helpful in improving the quality of the paper. We are more than encouraged that reviewers find:

+ our research topic of utilizing discriminative semantic priors within diffusion models to enhance discriminative task models to be **novel**, **interesting** and **valuable** (*Reviewer oxUu*).
+ our method to be with **significant advantage in efficiency** and **intriguing discovery** (*Reviewer V2Wk*), **novel** (*Reviewer Your*, *Reviewer oxUu*) and **valuable** (*Reviewer oxUu*), **well-motivated** (*Reviewer 6Z3D*).
+ our experimental results to show **remarkable gain** (*Reviewer V2Wk*), **outperform by large margins** (*Reviewer 6Z3D*, *Reviewer oxUu*) and **significant** (*Reviewer oxUu*), **adequate ablations** (*Reviewer 6Z3D*), **outstanding** and **reproducible** (*Reviewer Your*).
+ our advancements to be **particularly crucial** (*Reviewer V2Wk*).
+ our paper to be **well-written** (*Reviewer 6Z3D*), with **outstanding** and **well-designed** illustration (*Reviewer Your*).

We highly value the concerns and suggestions from each reviewer and have done our utmost to address them with detailed responses. A one-page PDF is also uploaded to provide better demonstrations of our responses, where the figures and tables are referred to as **Figure R** and **Table R** for the sake of clarity.

We hope that our rebuttal has sufficiently addressed the concerns raised by the reviewers. Please reply if you have any further questions, and we will be more than happy to continue the discussion.

---

### Decision · Program_Chairs · 2024-09-25

**Decision:**

Accept (poster)

**Comment:**

The reviewers acknowledge the soundness and the technical novelty of the paper. The rebuttal has a positive impact on the final ratings and all the reviewers scored weak accept or higher. The ACs followed the reviewers' recommendation.